# INSTANCE–SPECIFIC AUGMENTATION: CAPTURING LOCAL INVARIANCES

## ABSTRACT

We introduce *InstaAug*, a method for automatically learning input-specific augmentations from data. Previous data augmentation methods have generally assumed independence between the original input and the transformation applied to that input. This can be highly restrictive, as the invariances that the augmentations are based on are themselves often highly input dependent; e.g., we can change a leaf from green to yellow while maintaining its label, but not a lime. InstaAug instead allows for input dependency by introducing an *invariance module* that maps inputs to tailored transformation distributions. It can be simultaneously trained alongside the downstream model in a fully end-to-end manner, or separately learned for a pre-trained model. We empirically demonstrate that InstaAug learns meaningful input-dependent augmentations for a wide range of transformation classes, which in turn provides better performance on both supervised and self-supervised tasks.

## 1 INTRODUCTION

Data augmentation is an important tool in deep learning (Shorten & Khoshgoftaar, 2019). It allows one to incorporate inductive biases and invariances into models (Chen et al., 2019; Lyle et al., 2020), providing a highly effective regularization technique that aids generalization (Goodfellow et al., 2016). It has proved particularly successful for computer vision tasks, forming an essential component of many modern supervised (Perez & Wang, 2017; Krizhevsky et al., 2012; Cubuk et al., 2020; Mikołajczyk & Grochowski, 2018) and self-supervised (Bachman et al., 2019; Chen et al., 2020; Tian et al., 2020; Foster et al., 2021) approaches.

Algorithmically, data augmentations apply a *random transformation* $\tau : \mathcal{X} \to \mathcal{X}, \tau \sim p(\tau)$, to each input data point $\mathbf{x} \in \mathcal{X}$, before feeding this *augmented* data into the downstream model. These transformations are resampled each time the data point is used (e.g. at each training epoch), effectively populating the training set with additional samples. Augmentation is also sometimes used at test time by ensembling predictions from multiple transformations of the input. A particular augmentation is defined by the choice of the *transformation distribution* $p(\tau)$, whose construction thus forms the key design choice. Good transformation distributions induce substantial and wide-ranging changes to the input, while preserving the information relevant to the task at hand.

Data augmentation necessarily relies on exploiting problem-specific expertise: though aspects of $p(\tau)$ can be learned from data (Benton et al., 2020), trying to learn $p(\tau)$ from the set of all possible transformation distributions is not only unrealistic, but actively at odds with the core motivations of introducing inductive biases and capturing invariances. One, therefore, restricts $\tau$ to transformations that reflect how we desire our model to generalize, such as cropping and color jitter for image data.

Current approaches (Cubuk et al., 2018; Lim et al., 2019; Benton et al., 2020) are generally limited to learning augmentations where the transformation is sampled independently from the input it is applied to, such that $p(\tau)$ has no dependence on $\mathbf{x}$. This means that they are only able to learn *global invariances*, severely limiting their flexibility and potential impact. For example, when using color jittering, changing the color of a leaf from yellow to green would likely preserve its label, but the same transformation would change a lemon to a lime (see Figure 1b). This transformation cannot be usefully applied as a global augmentation, even though it is a useful invariance for the *specific input instance* of a leaf. Similar examples regularly occur for other transformations, as shown in Figure 1.

To address this shortfall, we introduce InstaAug, a new approach that allows one to learn *instance-specific* augmentations that encapsulate *local* invariances of the underlying data generating process,

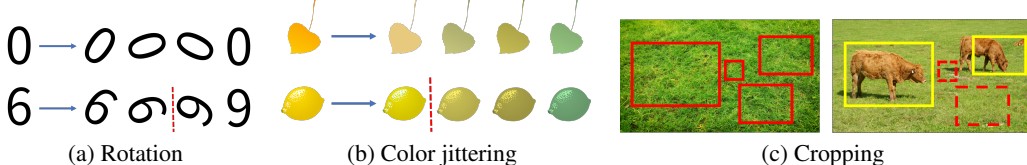

| (a) Rotation | (b) Color jittering | (c) Cropping |

Figure 1: Different inputs require different augmentations. In (a), the digit '0' is invariant to any rotation, but rotating the digit '6' by more 90° makes it a '9'. In (b), a similar phenomenon is observed for color jittering applied to a leaf and a lemon/lime. The red dashed lines in (a) and (b) are boundaries between different classes. In (c), the same effect is shown for cropping. Solid rectangles represent the patches that preserve the labels of the original images ([left] grass, [right] cattle), while dashed rectangles represent patches with different labels to the original images.

that is invariances specific to a particular region of the input space. InstaAug is based on using a transformation distribution of the form $p(\tau; \phi(\mathbf{x}))$, where $\phi$ is a deep neural network that maps inputs to transformation distribution parameters. We refer to $\phi$ as an *invariance module*. It can be trained simultaneously with the downstream model in a fully end-to-end manner, or using a fixed pre-trained model. Both cases only require access to training data and a single objective function that minimizes the training error while maintaining augmentation diversity. As such, *InstaAug* allows one to directly learn powerful and general augmentations, without requiring access to additional annotations.

We evaluate InstaAug in both supervised and self-supervised settings, focusing on image classification and contrastive learning respectively. Our experimental results show that InstaAug is able to uncover meaningful invariances that are consistent with human cognition, and improve model performance for various tasks compared with global augmentations. While we primarily focus on the case where the invariance module is trained alongside the downstream model (to allow data augmentation during training), we find that InstaAug can also provide substantial performance gains when used as a mechanism for learning test-time augmentations for large pre-trained models.

## 2 BACKGROUND

Data augmentation methods operate as a wrapper algorithm around some downstream model, $f$, randomly transforming the inputs $\mathbf{x} \in \mathcal{X}$ before they are passed to the model. The outputs of the augmented model are given by $f(\tau(\mathbf{x}))$, where $\tau : \mathcal{X} \mapsto \mathcal{X}$ represents the transformation, sampled from some transformation distribution $p(\tau)$. The aim of this augmentation is to instil inductive biases into the learned model, leading to improved generalization by capturing invariances of the problem. It can be used both during training to provide additional synthetic training data, and/or at test-time, where ensembling the predictions from multiple transformations can provide a useful regularization that often improves performance (Shanmugam et al., 2021).

Some approaches look to learn aspects of the augmentation (Cubuk et al., 2018; 2020; Lim et al., 2019; Ho et al., 2019; Hataya et al., 2020; Li et al., 2020; Zheng et al., 2022). These approaches can be viewed as learning parameters of $p(\tau)$, helping to automate its construction and tuning. Of particular relevance, Augerino (Benton et al., 2020) provides a mechanism for learning augmentations using a simple end-to-end training scheme, where the parameters of the downstream model and transformation distribution are learned simultaneously using the (empirical) risk minimization

$$\min_{f,\theta} \quad \mathbb{E}_{\mathbf{x},y \sim p_{\text{data}}} \left[ \mathbb{E}_{\tau \sim p_\theta(\tau)} \left[ \mathcal{L}(f(\tau(\mathbf{x})), y) \right] \right] + \lambda \mathbb{R}(\theta), \tag{1}$$

where $\mathcal{L}$ is a loss function and $\lambda \mathbb{R}(\theta)$ is a regularization term that encourages large transformations.

All of these approaches can be thought of as *global* augmentation schemes, in that transformations are sampled independently to the input. For an unrestricted, universal, class of transformations, this assumption can be justified through the noise outsourcing lemma (Kallenberg & Kallenberg, 1997): any conditional distribution $Y|X = x$ can be expressed as a deterministic function $g : \mathcal{X} \times \mathbb{R}^n \to \mathcal{Y}$ of the input and some independent noise $\varepsilon \sim \mathcal{N}(0, I)$. Thus, using reparameterization, the dependency on $\mathbf{x}$ can, in principle, be entirely dealt with by the transformation itself. However, in practice, the transformation class must be restricted to provide the desired inductive biases, meaning this result no longer holds and so the independence assumption can cause severe restrictions. For example, sampling rotations independently to the input is equivalent to the unrealistic assumption that the labels of all images $\mathbf{x}$ are invariant to the same range of angles (*cf.* Figure 1a).

## 3 INSTANCE-SPECIFIC AUGMENTATION: CAPTURING LOCAL INVARIANCES

In order to remedy the problems of global augmentations, we propose InstaAug. InstaAug learns an *input dependent* distribution $p(\tau; \phi(\mathbf{x}))$ of information-preserving transformations that actively makes use of the input $\mathbf{x}$ via the *invariance module* $\phi$, as opposed to learning a global transformation distribution $p_\theta(\tau)$. This generalizes the hypothesis class of transformation distributions, and significantly increases the flexibility and expressivity of the augmentations we can learn, without undermining our ability to carefully control the inductive biases that are imparted. It can also informally be viewed as a mechanism for learning invariances which are local to the specific input.

We argue that a good augmentation strategy needs to fulfill two properties. First, the transformations should preserve the information in $\mathbf{x}$ that is necessary for the task at hand. For example, in classification, transformations must preserve sufficient information to correctly classify $\tau(\mathbf{x})$. Second, the set of transformations needs to have sufficient 'diversity' to effectively augment the data; we quantify this as the entropy of the transformation distribution $p(\tau; \phi(\mathbf{x}))$. In addition to their intuitive nature, in Appendix A we provide theoretical analysis that shows these requirements naturally originate from a decomposition of the generalization error that results from using $\phi$ when training $f$. For simplicity, we describe InstaAug on the task of classification in the remainder of this section.

### 3.1 MODEL STRUCTURE

InstaAug is based around using a simple plug-in invariance module, $\phi$, between the input $\mathbf{x}$ and the classifier $f$, as shown in Figure 2. We assume a parametric family of distributions $p(\tau; \cdot)$ over some transformation space, then use $\phi$, which is a trainable neural network, to predicts its parameters for a given input. During training, we *sample* a transformation $\tau \sim p(\tau; \phi(\mathbf{x}))$, which is applied to $\mathbf{x}$ to generate an augmented sample $\tau(\mathbf{x})$, before feeding this into the classifier $f$.

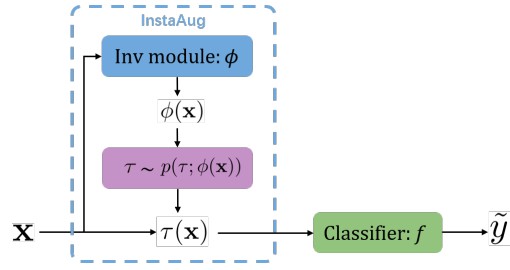

Figure 2: Summary of InstaAug.

### 3.2 TRAINING

Good augmentations should induce substantial changes to the input $\mathbf{x}$ while providing all necessary information of the task at hand, thereby capturing the maximum possible invariance. Figure 3a illustrates the tension between these two objectives experienced by global augmentation schemes. Wider-ranging transformations are generally beneficial for generalization, but 'excessive' transformations will generate samples that will be incorrectly classified. In Figure 3a we see this in the red area, where the augmentations for a pair of data points have started to overlap, creating ambiguity and inevitably misclassifications. Using instance-specific augmentations (Figure 3b) allows for a better trade-off of these needs. However, to achieve this we need our objective to encourage diversity in augmentations, not

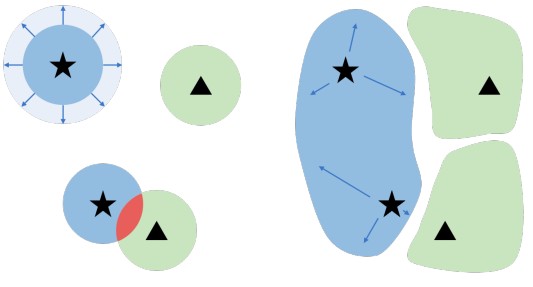

(a) Global augmentation     (b) InstaAug

Figure 3: InstaAug learns more diverse augmentations that also preserve labels compared to global augmentations. ★ and ▲ are samples from two different classes. Blue and green shades represent label-preserving augmentations for each class. In (a), the upper ★ should be further augmented, but some of the augmented samples for the lower ★ are already over-augmented and indistinguishable from another class (see the red intersection). InstaAug solves this problem by learning a different augmentation for each instance, as shown in (b).

just low training error. It should also let the level of diversity vary between inputs, as some points will be able to support larger transformations than others.

Based on these needs, training is done by simultaneously minimizing a conventional expected loss with respect to both $\phi$ and $f$ (or just $\phi$ if $f$ is a fixed pre-trained classifier as per Section 5.3), while placing a hard constraint on the average entropy of the transformations, $\mathbb{E}_{\mathbf{x} \sim p_{\text{data}}} [\mathbb{H}[p(\tau; \phi(\mathbf{x}))]]$.

The core motivation for this setup is that minimizing the expected loss will naturally encourage the information needed for prediction to be preserved, but the constraints on the entropy are needed to enforce diversity. Further motivation is provided by the theoretical analysis of Appendix A.

By appropriately parameterizing $p(\tau; \phi(\mathbf{x}))$ (see Section 3.3), we can write down its entropy in closed form. We can then formulate the problem as the following constrained optimization problem:

$$\min_{f,\phi} \quad \mathbb{E}_{\mathbf{x},y\sim p_{\text{data}}} \left[ \mathbb{E}_{\tau\sim p(\tau;\phi(\mathbf{x}))} \left[ \mathcal{L}(f(\tau(\mathbf{x})), y) \right] \right], \tag{2a}$$

$$\text{s.t.} \quad \mathbb{E}_{\mathbf{x},y\sim p_{\text{data}}} \left[ \mathbb{H}[p(\tau; \phi(\mathbf{x}))] \right] \in [\text{H}_{\min}, \text{H}_{\max}], \tag{2b}$$

where $\mathcal{L}$ is the loss, for which we will generally use the cross-entropy. Here the lower bound $p(\tau; \phi(\mathbf{x}))$ enforces the desired diversity. We typically expect this constraint to be active at the true optimal solution, so $\text{H}_{\min}$ can be thought of as a hyperparameter that controls the desired level of diversity. The upper bound prevents $p(\tau; \phi(\mathbf{x}))$ exploding at the start of training when the classifier is weak: without this we empirically find that the augmented samples from different classes tend to overlap in the initial phase of training, hindering the training of $f$.

The Lagrangian function, $\mathbb{E}_{\mathbf{x},y\sim p_{\text{data}}} \left[ \mathbb{E}_{\tau\sim p(\tau;\phi(\mathbf{x}))} \left[ \mathcal{L}(f(\tau(\mathbf{x})), y) \right] \right] - \lambda \mathbb{E}_{\mathbf{x},y\sim p_{\text{data}}} \left[ \mathbb{H}[p(\tau; \phi(\mathbf{x}))] \right]$, can be used to solve this constrained optimization, where $\lambda$ is the Lagrangian multiplier. In practice, we initialize $\lambda$ with a small positive value, and increase (decrease) $\lambda$ when the average entropy drops below $\text{H}_{\min}$ (exceeds $\text{H}_{\max}$). The invariance module and downstream model can thus be trained simultaneously using end-to-end gradient descent, utilizing the reparameterization trick to deal with the stochasticity of $\tau$ when possible (Kingma & Welling, 2014), and the REINFORCE estimator (Williams, 1992) otherwise. The approach can also be extended to regression or self-supervised learning by substituting the loss function $\mathcal{L}$ (*cf.* Appendix C).

### 3.3 PARAMETERIZATION OF AUGMENTATIONS

We focus on parameterizing transformations that are frequently used in computer vision, though our framework can easily be extended to other domains. Due to the varied characteristics of different image transformations, we design two different parameterization methods for $p(\tau; \phi(\mathbf{x}))$.

**Uniform parameterization.** For rotation and color jittering, we find that a uniform distribution is suitable for parameterizing $p(\tau; \phi(\mathbf{x}))$, such that $\phi(\mathbf{x})$ returns a pair $(\theta_{\min}, \theta_{\max})$ representing extrema of the possible transformations. For example, for rotations these represent the maximum and minimum rotation angles, such that $\tau(\mathbf{x}) = R(\theta) \cdot \mathbf{x}$, where $\theta \sim \mathcal{U}(\theta_{\min}, \theta_{\max})$. To compose multiple transformations (such as hue, saturation and brightness in color jittering), we simply sample them independently, such that $p(\tau_1, \ldots, \tau_K; \phi(\mathbf{x})) = \prod_{k=1}^{K} p(\tau_k; \phi_k(\mathbf{x}))$. This provides a similar parameterization to (Benton et al., 2020), but where $(\theta_{\min}, \theta_{\max})$ now critically varies with the input $\mathbf{x}$ and there is no symmetry assumption on the transformation ranges.

**Location-related parameterization.** Using this uniform parameterization is unfortunately not appropriate for cropping. Firstly, the distribution on crop centers may be multi-modal, since important information may exist in different parts of an image. Secondly, the desired crop size and center are often highly correlated so cannot be sampled independently. Finally, we encountered significant practical training issues when using the uniform parameterization for cropping, with $\phi$ often becoming trapped in local optima with little transformation diversity.

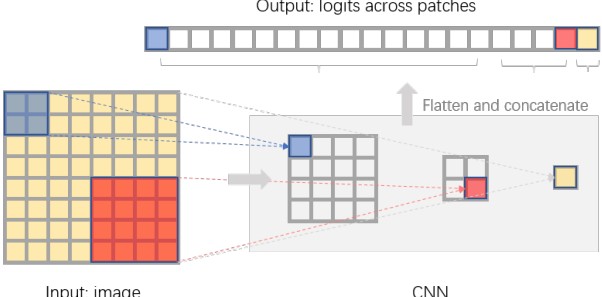

Figure 4: Location-related parameterization of crops by a CNN. The shaded area (bottom right) shows a simplified 3-layer CNN, and squares represent units at different convolutional layers. Each units defines a patch in the input image (shown in the same color) through its receptive field. The value of the activation then gives the corresponding unnormalized log probability for that patch.

We therefore propose an alternative location-related parameterization (LRP) for cropping, which is based on defining a large fixed set of allowable crops then constructing $\phi$ to map from inputs to a vector of probabilities over this set. As shown in Figure 4, this is achieved using

a CNN where each hidden unit corresponds to a one possible crop. The units from all layers are utilized, with those of earlier layers representing smaller crops. This parametrization proved more effective than simply outputting the probabilities from a conventional network, due to the greater parameter sharing between related crops. We note that it can also be directly extended to other transformations, such as masking, local blurring, pixel-wise perturbation, and local color jittering.

## 3.4 TEST-TIME AUGMENTATION

Besides augmenting data during training, the learned invariance can also be applied to test-time augmentation. Given a test image $\mathbf{x}$, we sample $n$ different transformations $\tau_i$ from $p(\tau; \phi(\mathbf{x}))$ and apply them to $\mathbf{x}$ to generate $n$ different views $\tau_i(\mathbf{x})$. After feeding these views to the classifier, $f$, we use the mean logit $\frac{1}{n} \sum_{i=1}^n f(\tau_i(\mathbf{x}))$ to predict $\mathbf{x}$'s label. When only learning invariance for test-time augmentation, InstaAug can be trained with a fixed pre-trained classifier at a lower computation cost.

## 4 RELATED WORK

**Hard-coded invariance.** Much recent work has been devoted to hard-coding global invariance in neural networks. For example, various models have been designed to be invariant to translation (Chaman & Dokmanic, 2021; Zhang, 2019), rotation (Worrall et al., 2017; Zhou et al., 2017; Marcos et al., 2017), scaling (Worrall & Welling, 2019; Sosnovik et al., 2019) or other group actions (Cohen & Welling, 2016; Xu et al., 2021). Unfortunately, they require the set of invariant transformations to be closed under composition, leaving out many practical transformations that do not form a group.

**Learning augmentations.** There have been numerous prior works that automatically learn *global* augmentations and invariance from data. As discussed in Section 2, Augerino (Benton et al., 2020) is perhaps the mostly closely linked such approach to InstaAug as it also relies on end-to-end training (see Appendix B for further discussion on its similarities and differences to InstaAug). AutoAugment (Cubuk et al., 2018) instead uses reinforcement learning to find augmentation strategies that increase accuracy on a separate validation set. Various follow-up works have improved its efficiency and/or performance (Lim et al., 2019; Ho et al., 2019; Hataya et al., 2020; Li et al., 2020; Cubuk et al., 2020; Tang et al., 2019; Zheng et al., 2022). A small number of works have further looked to learn augmentation policies that have some dependency on the input or just the class label (Zhou et al., 2021; Cheung & Yeung, 2022). These approaches focus on choosing *which type(s)* of transformation to apply from a fixed list—e.g. choosing from crop, blur, or color jitter—which may include a small number of discrete options for transformation strength. By comparison, InstaAug keeps the type of transformation fixed and learns instance-specific parameter for the transform distribution, such as the positions and sizes of patches for cropping. In principle, it should be possible to combine these complementary approaches with InstaAug, though we note they require a separate validation dataset and cannot be used in unsupervised settings, unlike InstaAug.

**Other related work.** The spatial transformer (Jaderberg et al., 2015) aims to learn instance-specific transformations, but only applies a single transformation to each input rather than a distribution of transformations, making it distinct from data augmentation. Luo et al. (2020) and Kim et al. (2020) both also learn instance-specific augmentations. However, the latter consider only test-time augmentation, while the former introduces an approach that is highly specialized to test recognition and cannot be applied in more general settings we consider. Tamkin et al. (2020) and (Chen et al., 2021) both utilize adversarial augmentations to increase robustness. Zhou et al. (2020) learn symmetries shared across several datasets through a meta-learning scheme.

## 5 SUPERVISED LEARNING EXPERIMENTS

### 5.1 ROTATED 2D IMAGES

We first consider a toy synthetic dataset proposed in Benton et al. (2020). The dataset contains four categories, (1) upright Mario; (2) upside-down Mario; (3) upright Iggy; and (4) upside-down Iggy. Each of the four base images is randomly rotated in the interval of $[-\pi/4, \pi/4]$ to form the training dataset. The task is to predict the correct character (Mario vs Iggy) and the orientation (up vs down). We assess whether InstaAug is able to learn the 'best' rotation range for each sample—i.e. the maximum range that avoids 'up' and 'down' classes from overlapping.

Figure 5 shows that InstaAug effectively recovers the broadest range of rotations for each image while preserving labels, while Augerino only learns a subset of these ranges. This can be most easily seen by the fact that the transformation distributions (shown in green) always extend to very close to the true class boundary for InstaAug, but not for Augerino. These gains are because Augerino learns a *single* global augmentation distribution shared across all images (note the shared transformation distribution arcs), which are inevitably limited for any given input.

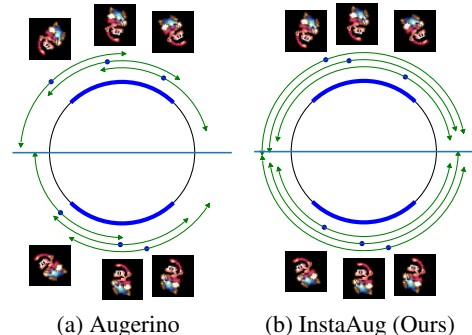

(a) Augerino    (b) InstaAug (Ours)

Figure 5: Learned invariances for the Mario and Iggy dataset. The blue arcs show the training data range, while the green arcs show the learned transformation distributions for some examples.

## 5.2 CROPPING

We now move to more realistic images and to the most common and effective form of image augmentation: cropping. We first evaluate the performance of jointly training InstaAug and the classifier on Tiny-Imagenet (TinyIN, $64 \times 64$), as it inherits the image complexity of ImageNet whilst being within our computational budget. TinyIN is a standard testbed for data augmentations that contains 100k images divided into 200 classes. Full experiment details are given in Appendix D.1.

We benchmark InstaAug alongside several augmentation baselines, including Augerino, no augmentation, and random crops (random augmentation). The latter uniformly samples patch sizes and then randomly selects a patch inside the image. Since the effect of cropping crucially relies on scales of patches, we carefully tune this baseline by sweeping over all possible scale intervals between $[0, 1]$ with a stride of $0.1$. We further compare to other prior works that have obtained competitive results on TinyIN (Ramé et al., 2021; Yun et al., 2019; Zhang et al., 2018).

In order to ablate the effects of input-dependency and location-related parameterization on InstaAug, we additionally assess the performance of *InstaAug (without LRP)* which relies on the same uniform parameterization as Augerino rather than our location-related parametrization (LRP, described in Figure 4); *InstaAug (without input)* that uses the LRP and general InstaAug setup, but shares the transformation distribution across all inputs rather than learning an input-specific augmentation; and *InstaAug (class specific)*, which takes training labels instead of images as inputs. Test-time augmentation using 50 transformation samples is deployed for all variants of InstaAug, along with the Augerino and random augmentation baselines. For InstaAug (class specific), this test-time augmentation is based on random cropping,

Table 1: InstaAug improves generalization on Tiny-ImageNet by learning instance-specific cropping. 'Instance' and 'LRP' refer respectively to 'instance-specific' and 'location-related parameterization'. Statistics are computed over 10 runs, except for MixMo, CutMix and Mixup, whose results are from Ramé et al. (2021). Other learnable augmentation methods are actually learning the size ranges of cropping. We leave their results out because we are already performing this learning in the random cropping results through our hyperparameter tuning.

| Method | Instance | LRP | Accuracy (%) |
|---|---|---|---|
| MixMo (Ramé et al., 2021) | — | — | 64.80 |
| CutMix (Yun et al., 2019) | — | — | 65.09 |
| Mixup (Zhang et al., 2018) | — | — | 63.74 |
| No augmentation | ✗ | ✗ | $55.06_{\pm 0.10}$ |
| Random crop | ✗ | ✗ | $64.49_{\pm 0.12}$ |
| Augerino (Benton et al., 2020) | ✗ | ✗ | $55.02_{\pm 0.29}$ |
| InstaAug (without LRP) | ✓ | ✗ | $55.39_{\pm 0.19}$ |
| InstaAug (without input) | ✗ | ✓ | $63.20_{\pm 0.12}$ |
| InstaAug (class specific) | ✗/✓ | ✓ | $60.55_{\pm 0.50}$ |
| InstaAug | ✓ | ✓ | $\mathbf{66.02}_{\pm 0.18}$ |

due to the lack of class information being available at test-time and this approach performing better than simply omitting test-time augmentation. Following prior works, we choose the PreActResNet-18 architecture (He et al., 2016b) with width = 1 as the classifier for all methods.

Table 1 shows the top-1 accuracy for each method. In agreement with prior works, we find that random cropping increases top-1 accuracy by 9.4% over no augmentation, which is achieved where cropping scale = $[0.1, 1]$. InstaAug outperforms random cropping and its own global version without input by 1.5% and 2.8% respectively, highlighting the effect of learning instance-specific augmentation.

Allowing only for class dependence actually produces even worse performance than just ignoring the input completely, presumably because of the inevitable resulting mismatch in the augmentations used in training and testing. Methods with mean-field uniform parameterization (including Augerino and InstaAug without LRP) performed extremely poorly, noticeably worse than just random cropping. This is because they were found to become easily stuck at local minima with low cropping diversity, leading to similar performance as *no augmentation*. Note that the potentially unexpectedly good performance of the random cropping baseline compared to the other global baselines stems from the careful hyperparameter sweep used to tune its crop size, which proved more effective than these more direct training mechanisms. See Appendix E.3 for more discussion.

Figure 6 shows example crops and learned transformation distributions for InstaAug and a global augmentation scheme (InstaAug without input). We see that InstaAug is able to learn a cropping scheme that focuses on the key aspect of the input image, while the baselines cannot.

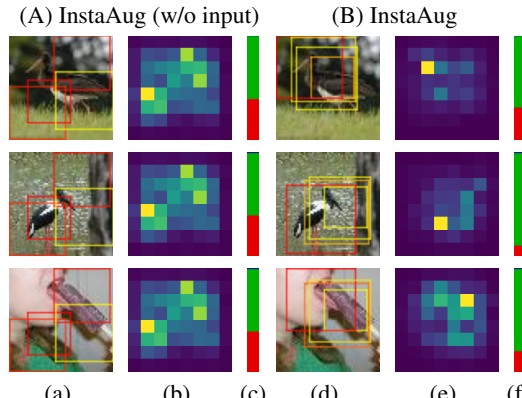

Figure 6: InstaAug (B) learns more sensible crops compared to random and learned global (A) augmentations. Columns (a, d) show examples of sampled crops, with red edges indicating higher probability. Columns (b, e) show density maps for the crop centres, with brighter color meaning higher probability. Columns (c, f) give the proportion of crops (red) above a particular size threshold, showing that InstaAug produces fewer large crops.

## 5.3 APPLYING INSTAAUG TO A FIXED CLASSIFIER

InstaAug can also be used to learn suitable augmentations for a fixed pre-trained classifier. This can most notably be useful as a means to learn test-time augmentations. As the invariance module is itself only a small network, it can be done relatively cheaply, even when the dataset and downstream model are very large. We exploit this on the larger Imagenet dataset ($224 \times 224$) (Deng et al., 2009), again focusing on cropping augmentations and utilizing the LRP parameterization from Section 3.3.

Training the invariance module in this setting is done in exactly the same way as elsewhere, using the training procedure of Section 3.2 with the normal training data. The only thing that is changed is that $f$ is now fixed to a pre-trained classifier—specifically, the ResNet-50 (He et al., 2016a) from Wightman (2019) (which did not use an invariance module during training)—rather than being simultaneously learned. We are thus simply learning invariances, without affecting the training of $f$.

In Table 2 we show the effect of using the learned invariance module for test-time augmentation, finding that it is able to noticeably improve accuracy, unlike the baseline test-time augmentations of random cropping, AutoAugment (Cubuk et al., 2018), and Fast AutoAugment (Lim et al., 2019). In order to evaluate the generalization performance of our learned augmentation module, we further apply the augmentation trained on ResNet-50 to two different models *with zero fine-tuning*: ResNet-18 (He et al., 2016a) and XCiT (Ali et al., 2021). We find that the learned augmentation trans-

Table 2: InstaAug boosts the test accuracy (%) with test-time augmentation on Imagenet. Invariance modules learned on ResNet-50 can also be directly applied to other models such as ResNet-18 and XCiT to improve generalization without fine-tuning. By contrast, we see that global augmentation schemes are actually detrimental to test-time augmentation.

| Method | #Sample | ResNet50 | ResNet18 | XCiT |
|---|---|---|---|---|
| No aug | 1 | 80.43 | 69.73 | 86.34 |
| Random crop | 4 | 78.45±0.04 | 66.13±0.04 | 82.05±0.01 |
| AutoAug | 4 | 77.84±0.05 | 59.50±0.01 | 81.40±0.00 |
| FastAutoAug | 4 | 77.87±0.06 | 61.43±0.02 | 81.42±0.01 |
| InstaAug | 4 | **80.92**±0.04 | **70.59**±0.05 | **86.43**±0.04 |
| Random crop | 10 | 79.60±0.01 | 67.87±0.01 | 82.84±0.00 |
| AutoAug | 10 | 79.20±0.04 | 63.96±0.03 | 82.43±0.02 |
| FastAutoAug | 10 | 79.28±0.01 | 65.65±0.02 | 82.45±0.02 |
| InstaAug | 10 | **81.18**±0.02 | **70.96**±0.03 | **86.47**±0.02 |

fers very effectively to these different models, which implies that the local invariances InstaAug learns reflect the natural invariances of the underlying classification problem, rather than being specific to the model that was used to train the augmentation module.

## 5.4 COLOR JITTERING ON TEXTURES

Color jittering is another important type of data augmentation, which can help models generalize to different lighting conditions. We benchmark on the texture classification dataset RawFooT (Bianco et al., 2017). RawFooT includes 68 different samples of raw food and each sample has an image taken under each of 46 different lighting conditions (see Figure D.1 for some examples). We crop the original images to create the train set and test set. For each original image with a resolution of $800 \times 800$, we randomly sample 200 different $200 \times 200$ patches in the upper half as training images. The same procedure is taken on the lower half to produce test images, giving a train set and a test set for each different lighting condition. To evaluate the generalization ability of each method to a broader range of lighting conditions, we evenly mix test images from all lighting conditions to form a general test set, while controlling the lighting conditions present during training.

We first train on a single lighting condition D45 (4500K, daylight) resembling natural light. Table 3 shows that InstaAug outperforms all baselines with and without test-time augmentation. In this task, we find that Augerino (with relaxed symmetry restrictions on learned intervals) underperforms random augmentation because its parameters $\phi$ are often stuck in a neighborhood around their initial values. We believe this is due to the conservative nature of using global augmentations (*cf.* Figure 3), where even a small change in the parameters may largely increase the training loss, which prohibits wide-ranging augmentations.

Table 3: InstaAug achieves higher general accuracy than baseline methods when trained on D45 (Daylight, 4500K).

| Method | Test aug? | Accuracy (%) |
|---|---|---|
| No aug | ✗ | $72.87_{\pm 0.10}$ |
| Random aug | ✗ | $79.99_{\pm 0.13}$ |
| Augerino | ✗ | $78.97_{\pm 0.10}$ |
| InstaAug | ✗ | $\mathbf{81.11}_{\pm 0.20}$ |
| Random aug | ✓ | $80.55_{\pm 0.16}$ |
| Augerino | ✓ | $79.34_{\pm 0.14}$ |
| InstaAug | ✓ | $\mathbf{81.35}_{\pm 0.19}$ |

We also compare in-distribution and out-of-distribution generalization by splitting the 46 test sets into two groups, according to the similarity of their lighting conditions to D45—see Appendix D.2 for the details on the splitting method. In Figure 7 we can see that above a certain in-distribution performance, there exists a trade-off for random augmentation between in-distribution accuracy and out-of-distribution generalization, controlled through the hyperparameter settings. InstaAug, meanwhile, delivers higher out-of-distribution performance than any of the hyperparameter configurations, while also simultaneously giving better in-distribution accuracy to the vast majority of them as well.

We can further vary the difficulty of the classification task by using different numbers of lighting conditions in the training data. In Table 4, we randomly select a set number of lighting conditions to use as the training set for each baseline. As expected, the accuracy increases with the number of lighting conditions for all meth-

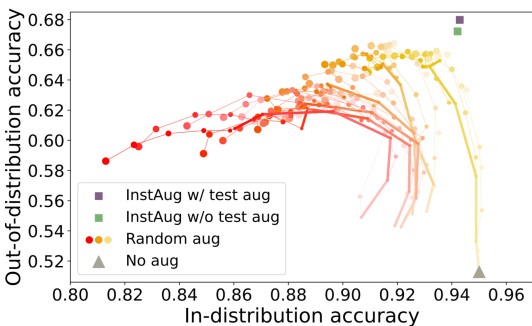

Figure 7: In-distribution and out-of-distribution test accuracy for models trained on RawFooT D45. The round dots are random augmentation with different hyperparameter settings. The colors of dots change from yellow to red as hue jittering increases; more saturated dots indicate higher saturation jittering; larger dots mean higher brightness jittering. Each thick line connects dots with the same hue and brightness jitter and thin lines link dots with the same hue and saturation jitter.

ods. However, the effect of random augmentation saturates: it performs similarly to no augmentation with 8 lighting conditions. By contrast, InstaAug always provides improvements. In Appendix D, we show that these gains come at very little computational overhead at both train and test time.

## 6 INSTAAUG FOR CONTRASTIVE LEARNING

Contrastive learning aims to learn features that are approximately invariant to certain augmentations. Typical contrastive learning methods, such as SimCLR (Chen et al., 2020; Ermolov et al., 2021), first sample two independent transformations, $\tau_1, \tau_2 \sim p(\tau)$, and apply them to an input image $\mathbf{x}$,

Table 4: InstaAug significantly outperforms baseline methods in general test accuracy (%) on different difficulty levels. For each difficulty level, we randomly sample lighting conditions used for training and repeat each experiment 10 times. Test-time augmentation is included for random and InstaAug.

| Method / #Lighting conditions for training | 1 | 2 | 4 | 8 |
|---|---|---|---|---|
| No aug | $68.5_{\pm2.6}$ | $78.1_{\pm1.8}$ | $84.8_{\pm0.7}$ | $87.8_{\pm0.5}$ |
| Random augmentation | $72.7_{\pm2.7}$ | $80.8_{\pm1.3}$ | $85.9_{\pm0.6}$ | $87.3_{\pm0.3}$ |
| InstaAug | $\mathbf{76.0}_{\pm2.5}$ | $\mathbf{83.6}_{\pm1.1}$ | $\mathbf{88.2}_{\pm0.5}$ | $\mathbf{89.6}_{\pm0.3}$ |

generating two views $x_1$ and $x_2$. They then feed the transformed images to a neural encoder $f$, which is trained to maximize the similarity between $f(x_1)$ and $f(x_2)$, measured with a contrastive loss.

As the choice of augmentations directly influences the learned invariance of the encoder, it is a crucial ingredient of contrastive learning (Bachman et al., 2019; Chen et al., 2020; Tian et al., 2020). However, existing schemes use global augmentations which often introduce unrealistic assumptions. For example, if there are multiple entities in an image, such as grass and cattle in Figure 1c, random cropping will pull features for different entities closer to each other. Consequently, we propose InstaAug as a more flexible instance-specific augmentation method for contrastive learning.

Applying InstaAug to contrastive learning is similar to the supervised case shown in Section 3. The main difference is, given an input $x$, we sample two $\tau$ independently from the input-specific distribution $p(\tau; \phi(x))$, before they are applied to $x$. The training objective is correspondingly changed to minimizing the contrastive loss while keeping the diversity in a reasonable range.

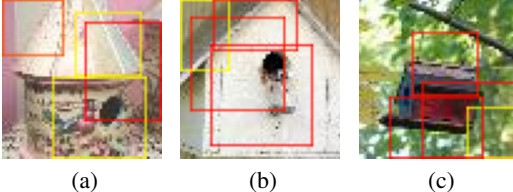

(a)          (b)          (c)

Figure 8: Some examples (bird houses) of learned cropping in contrastive learning.

We again consider TinyIN and evaluate three methods: InstaAug, InstaAug (without input), and random crop. We exclude methods with uniform parameterization, because of their poor performance in the simpler supervised setting. All experiments are based on the SimCLR framework and use the PreActResNet-18 network as the encoder. We train each model with a batch size of 512 for 500 epochs. We then train a linear classifier to evaluate feature quality. We use test-time augmentation—with 10 sampled crops—as it has been shown to improve performance (Foster et al., 2021).

From Table 5, we see that InstaAug outperforms the random and global augmentation schemes as well as Un-Mix (Shen et al., 2022), which is a recent variant of MixUp methods on contrastive learning. We observe from the examples shown in Figure 8 that InstaAug focuses on the salient features containing important information. We also notice that the sizes of learned patches are correlated to the sizes of the main objects in images. Thus, InstaAug is able to learn sensible instance-specific augmentations in a fully unsupervised setting.

Table 5: Representations learned by InstaAug perform better in the downstream linear classification task than baselines. *Results of Un-Mix are directly taken from Shen et al. (2022), which has the same network structure (ResNet-18), training algorithms (SimCLR) and linear classifier as ours.

| Method | Accuracy (%) |
|---|---|
| Un-Mix (Shen et al., 2022) | $49.58^*$ |
| Random crop | $51.63_{\pm0.30}$ |
| InstaAug (without input) | $54.20_{\pm0.23}$ |
| InstaAug | $\mathbf{55.05}_{\pm0.21}$ |

## 7 DISCUSSION

In this paper we introduced InstaAug, a method for learning instance-specific data augmentations that capture local invariances of the underlying data generating process. This is achieved by training an augmentation module that parametrizes an input-dependent distribution over transformations, whose samples are used to augment the training data on the fly and/or for test-time augmentation. The main benefits of InstaAug stem from its applicability to a wide range of settings, its ease of use, and crucially its capacity to learn meaningful augmentations that in turn improve performance. Empirically, we demonstrated these benefits for both classification and contrastive learning problems, considering several classes of transformations—rotation, color jittering, and cropping.

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

## APPENDIX A  THEORETICAL ANALYSIS OF GENERALIZATION ERROR

We now provide a decomposition of the generalization error—i.e. the difference between the true risk and the training risk—when using $\phi$ during training of the downstream classifier $f$. Here we can view the objective of augmentation as adjusting the training objective to encourage the learned model to have a low true risk. As such, the generalization error provides a measure of the effectiveness of the augmentation for the training of $f$; by analysing the behaviour of the generalization error as a function of the augmentation module, we can derive a characterization of the desirable properties of the latter.

To start our analysis, we first define the true risk of the downstream model, $f$, as

$$R(f) := \mathbb{E}[\mathcal{L}(f(X), Y)] \tag{A.1}$$

where $(X, Y) \sim p_{\text{true}}(X, Y)$ are drawn from the true data generating distribution. In practice, one might also perform test-time augmentation, implying a different predictive function and thus different true risk, but for the purposes of our analysis, we will assume that this is not done, as this allows us to focus on the impact the invariance module has on $f$ during training.

One the other hand, the implied training risk (i.e. our objective for training $f$) when using an invariance module is the augmented empirical risk

$$\hat{R}(f, \phi) := \mathbb{E}[\mathcal{L}(f(\tau(x_i)), y_i)] \tag{A.2}$$

where $i \sim \text{Uniform}\{1, \ldots, N\}$ is a uniformly sampled index for a point in the original training dataset $\{x_n, y_n\}_{n=1}^N$ and $\tau|i \sim p(\tau; \phi(x_i))$ is the sampled transformation. Note that the expectation in Equation (A.2) is only over $i$ and $\tau$, with the datapoints themselves not considered random variables for our purposes, because we are only provided with a single fixed training dataset.

The generalization error can now be defined as $\hat{R}(f, \phi) - R(f)$. At a high level, we are interested in finding a $\phi$ that ensures this has a low magnitude. More precisely, we want $\phi$ to ensure that the minimizer of the training risk, $\hat{f}^* := \arg\min_f \hat{R}(f, \phi)$, gives as low a true risk, $R(\hat{f}^*)$, as possible. Therefore, we want to keep the generalization error magnitude small across different $f$ (relative to the corresponding variations in $\hat{R}(f, \phi)$ itself), so that the optima of the training and true risks are as similar as possible. In other words, we want a $\phi$ that ensures $\hat{R}(f, \phi) - R(f)$ is small for *all* $f$, especially those close to $\hat{f}^*$. If we do hypothetically drive the generalization error to zero for all $f$, we will have a mechanism for directly training to the true risk using a finite original training dataset.

To aid with decomposing the generalization error, it is convenient to further define the following random variables through their conditional distributions:

$$\hat{Y}|i \sim p_{\text{true}}(Y = \hat{Y}|X = x_i) \quad \text{with} \quad \hat{Y} \perp\!\!\!\perp \tau, \tag{A.3}$$

$$\tilde{Y}|i, \tau \sim p_{\text{true}}(Y = \tilde{Y}|X = \tau(x_i)). \tag{A.4}$$

We can now write down our decomposition as follows:

$$\hat{R}(f, \phi) - R(f) = \underbrace{\mathbb{E}[\mathcal{L}(f(\tau(x_i)), \hat{Y}) - \mathcal{L}(f(\tau(x_i)), \tilde{Y})]}_{(A)}$$
$$+ \underbrace{\mathbb{E}[\mathcal{L}(f(\tau(x_i)), \tilde{Y}) - \mathcal{L}(f(X), Y)]}_{(B)} + \underbrace{\mathbb{E}[\mathcal{L}(f(\tau(x_i)), y_i) - \mathcal{L}(f(\tau(x_i)), \hat{Y})]}_{(C)}. \tag{A.5}$$

From this, we see that if the magnitude of (A), (B), and (C) are all small, then our generalization error magnitude will be small as well. Moreover, if we can construct a $\phi$ such that these terms are small for *all* $f$, then we can ensure effective generalization performance. We will now look at each term individually.

(A) provides a precise characterisation of how well our transformation preserves the label distribution; it is the difference between the expected loss under the true label distribution of the untransformed inputs and the expected loss under the true label distribution of the transformed inputs, making predictions using the transformed inputs in both cases. In particular, by noting that we have

$$(A) = \mathbb{E}\left[\mathbb{E}\left[\mathcal{L}(f(\tau(x_i)), \hat{Y}) - \mathcal{L}(f(\tau(x_i)), \tilde{Y})\Big|i, \tau\right]\right] \tag{A.6}$$

where $f(\tau(x_i))$ is deterministic given $\tau$ and $i$, we have that $\tilde{Y}|i,\tau, \overset{\mathrm{d}}{=} \hat{Y}|i, \forall i,\tau$ is a sufficient (but not necessary) condition to ensure (A) $= 0$ for all $f$.[1] That is, it is zero for all $f$ if the conditional distribution on the labels is the same for both the original and transformed inputs for all possible pairs $(i,\tau)$, i.e. all possible original inputs and sampled transformations. One simple way to ensure this is to have $\tau$ always be equal to the identity mapping, so this term prefers limited transformations.

By contrast, if the transformation destroys information about the label, $\hat{Y}|i$ and $\tilde{Y}|i,\tau$ will now differ, such that, in general, (A) $\neq 0$ and, moreover, it will vary with $f$. Here we typically expect that (A) $\geq 0$,[2] as we are making predictions using the transformed inputs, so the expected loss under the true label distribution for the transformed inputs will tend to be less than that when labels are generated using the untransformed input. To keep the magnitude of (A) low, we need to ensure that transformations maintain the conditional label distribution as well as possible, i.e. that transformations preserve all input information that is salient for predicting labels.

Conveniently, minimizing $\hat{R}(f,\phi)$ with respect to $\phi$, as done by the InstaAug training setup of Section 3.2, will naturally try to reduce (A). Given we expect the term to typically be positive, this provides an explanation for why InstaAug can be effective without any separate consideration in the objective for the need for transformations to maintain the class label distribution.

(B) represents how well our transformation captures the true input distribution. Here we can utilize the fact that, by the definition of $\tilde{Y}$,

$$\mathbb{E}\left[\mathcal{L}(f(\tau(x_i)),\tilde{Y})\Big|\tau(x_i)=x\right] = \mathbb{E}\left[\mathcal{L}(f(X),Y)|X=x\right] =: r(x) \tag{A.7}$$

to write it as

$$(B) = \mathbb{E}[r(\tau(x_i))] - \mathbb{E}[r(X)], \tag{A.8}$$

where $r : \mathcal{X} \mapsto \mathbb{R}^+$ maps inputs to their true expected loss. We thus see that $\tau(x_i) \overset{\mathrm{d}}{=} X$ is a sufficient (but not necessary) condition to ensure that (B) $= 0$ for all $f$. That is (B) is always $0$ if the process of choosing one of the training inputs at random followed by applying a sampled transformation to that input produces samples distributed exactly according to the true input distribution. Unlike for (A), there is no simple scenario in which we can ensure this is true, with the use of the identity transformation now likely to give significant discrepancies by failing to provide sufficient coverage of the input space: though the $x_i$ may originally have been sampled from $p_{\mathrm{true}}(X)$, there is only a finite set of them, such that repeated sampling from this finite set represents a substantially different distribution to $p_{\mathrm{true}}(X)$. In fact, (B) nicely encapsulates the desire to perform augmentation in the first place, by showing how it can be used to increase the coverage of the input space.

How to best manage Term (B) will vary depending on the type of model used and the form of our transformations. In some situations, it may be that no matter how diverse our transformations are within the class of those allowable, $\tau(x_i)$ will still only cover a subset of the support of $X$. Here the most important factor for keeping (B) small will be to maximize the diversity of the transformations, e.g. by maximizing their entropy, to ensure the best possible coverage of the true input space. In other cases, it might also be possible to "over–diversify" the inputs, such that $\tau(x_i)$ can become more diffuse than $X$ for some choices of $\phi$, potentially causing training to lack focus on the particular test-time input distribution we care about. Here we may need to ensure that the entropy of the transformation does not become so large as to cause such over-diversification, creating a more complex trade-off with the need to ensure sufficient coverage. These two scenarios respectively motivate the lower and upper bounds on the transformation distribution entropy used when training the augmentation module.[3]

For augmentation of high-dimensional data, the former, coverage-limited, scenario is expected to be significantly more likely, as our original training data will generally provide quite poor coverage

---

[1]Note that $\hat{Y} \overset{\mathrm{d}}{=} \tilde{Y}$ alone is not generally sufficient, as matching in marginal distribution does not ensure that the joint distributions with $i$ and $\tau$ also match, in turn yielding different expectations.

[2]Note, though, that this is not formally guaranteed, even for the cross entropy loss and an $f$ that exactly captures the true distribution. This is because, while Gibbs' inequality ensures the optimal $q$ given $p$ for a cross-validation expected loss $\mathbb{E}_{p(Y)}[-\log q(Y)]$ is $q = p$, in general, the optimal $p$ given $q$ is not $p = q$.

[3]Note here that the bounds in Equation (2b) are on are on the entropy on the parameters of $\tau$, rather than $\tau(x_i)$ itself. This is because it is difficult to directly control the latter during the training, with the former providing a more practical proxy that is expected to generally be representative.

of the true input distribution, while our transformations will not generally be sufficiently powerful to produce unrepresentative inputs. Moreover, when working with large deep learning models, prediction in one region of the input space is rarely harmed by the addition of data in another input region. Thus, for the typical scenarios, we expect InstaAug to be deployed in, increasing the entropy of the transformations will directly relate to reducing the magnitude of (B). Note here that it will typically be the case that (B) < 0 provided that the transformations maintain the label distribution, as the accuracy of the downstream model will typically be higher for the transformations of the original training data that for the test data.

Term (C) is the error from the fact that we only have one sample of the label for each original training input, rather than the full label distribution. As $\hat{Y} \perp\!\!\!\perp \tau$, we have limited ability to reduce it through controlling $\phi$; it essentially represents the irreducible noise in $\hat{R}(f, \phi)$ from only having a finite number of true labels. Note that it is not related to the model's ability to generalize to unseen inputs, as it is based on variability in other possible labels we might have seen for our training inputs themselves; if $Y|X$ is actually deterministic, it is exactly zero. As such, it is of limited interest for our analysis, while it will thankfully generally be much smaller than the other terms for practical problems unless we have both a very small dataset and a very noisy true label distribution.

Putting everything together, we see that (A) and (B) respectively encapsulate the competing needs of the invariance module to maintain the conditional label distribution (i.e. preserve the label information) and maximize coverage of the input space. We have also seen that the former is typically naturally taken care of by minimizing $\hat{R}(f, \phi)$ with respect to $\phi$, motivating the objective used by InstaAug in Equation (2a), but the latter requires separate consideration, which we deal with through our constraints on the entropy in Equation (2b).

## APPENDIX B   DETAILS OF AUGERINO

As a method to learn invariance, Augerino (Benton et al., 2020) is quite different from the previous approaches, which usually require an extra validation set. The basic idea behind Augerino is to use a few parameters ($\theta$) to control the transformation distribution on input images and learn these parameters with the training loss of the classifier. Specifically, it minimizes the loss

$$\mathcal{L}_\lambda(\mathbf{x}; y) \triangleq \mathbb{E}[\mathcal{L}(\mathbf{x}; y)] + \lambda \cdot \mathbb{R}(\theta), \tag{B.1a}$$

where $\mathcal{L}(\mathbf{x}; y)$ is the cross-entropy loss and $\mathbb{R}(\theta)$ is a regularization function on the volume of the support of the distribution weighted by the hyper-parameter $\lambda$.

**Comparison with InstaAug.**   InstaAug shares with Augerino the ideas of tuning augmentation parameters by the classifier loss and using test time augmentation to boost performance, but they are different in the following aspects. The most significant difference is that InstaAug is instance-specific, while Augerino learns global augmentations. Besides, Augerino uses a single scalar $\theta$ to parameterize a symmetric uniform distribution ($\mathcal{U}[-\theta, \theta]$) over each type of transformations, which lacks the flexibility to model more complex augmentations, such as cropping.

In addition, Augerino uses a fixed weight $\lambda$ to balance the training loss and augmentation diversity. However, we find that, in more complicated settings, this is quite impractical. Specifically, we need different $\lambda$ in different stages of training. If we use a large $\lambda$ from the start of training, the diversity will quickly diverge to maximum, because the classifier is very weak and the loss is consequently dominated by the diversity term. This will block the training of the classifier because transformed samples from different classes are quite mixed with each other. Otherwise, if we choose a small $\lambda$, the diversity will converge to zero after a few epochs, yielding similar results as the vanilla model without augmentation. In neither of the case can we learn a useful augmentation. Consequently, InstaAug directly constrains the diversity to keep it stable during training.

## APPENDIX C   METHOD DETAILS

### C.1   REGRESSION AND SELF-SUPERVISED LEARNING

In Section 3, we use classification as an example to introduce InstaAug. However, InstaAug can be easily applied to other tasks including regression and self-supervised learning. For regression,

the classifier (see Figure 2) is replaced by a regressor and the loss function $\mathcal{L}$ in Equation (2a) is changed accordingly to absolute or square error. For self-supervised contrastive learning, we replace the classifier and cross-entropy loss with the feature extractor and contrastive loss (such as SimCLR loss (Chen et al., 2020)), respectively. In addition, the sampler samples 2 rather than 1 transformations to generate multiple views for an input $\mathbf{x}$.

### C.2 Implementation of location-related parameterization

As an example, we show how to implement location-related parameterization with a basic CNN structure in the following algorithm,.

---

**Algorithm 1:** Location related parameterization

---

**Input:** Image $\mathbf{x}$, channel numbers $M_i$, and layer number $n\_layer$
**Output:** Probability of patches $\mathbf{p}$
$\mathbf{F}_0' = \mathbf{x}$;
**for** ( $i = 1$; $i \leq n\_layer$; $i = i + 1$ )
    $\mathbf{F_i} = \text{Conv2d}(\mathbf{F_{i-1}'}, \text{kernel=2}, \text{stride=1}, \text{output\_channel=}M_i)$ ;    `// CNN Operation`
    $\mathbf{F_i'} = \text{Pooling}(\mathbf{F_i}, \text{kernel=2})$ ;    `// CNN Operation`
    $\mathbf{F_i''} = \text{Conv2d}(\mathbf{F_i'}, \text{kernel=1}, \text{stride=1}, \text{output\_channel=1})$ ; `// To a single channel`
    $\mathbf{logit_i} = \text{Flatten}(\mathbf{F_i''})$ ;    `// Logit vector at each level`
$\mathbf{logits} = \text{Concat}([\mathbf{logit_i}])$ ;    `// Logit vector at all levels`
$\mathbf{p} = \text{Normalize}(\text{Exp}(\mathbf{logits}))$ ;    `// Probability after normalization`

---

### C.3 Other parametrization methods

Besides the uniform and location-related parameterization, we also tried VAE-like methods to parameterize augmentations, such as cropping. The main idea is to have a Gaussian latent variable and a neural decoder to map the latent Gaussian distributions to a continuous distribution on transformation parameters (in this case, the centers and sizes of crops). However, similar to the uniform parameterization, we find the VAE-like parameterization unstable and easily stuck at local minima.

## APPENDIX D   EXPERIMENTAL DETAILS

### D.1 Cropping

**Supervised training**    Based on the Mixmo codebase[4] (Ramé et al., 2021), we use stochastic gradient descent (SGD) optimizer to train baselines and InstaAug. For the classifier, the initial learning rate is set to $0.2$ (with momentum $0.9$ and weight decay $1e-4$). A scheduler is used to decrease the learning rate by a factor of $0.9$ once validation accuracy doesn't increase for 10 epochs. The learning rate of the augmentation module $\phi$ is fixed at $1e-5$. Batch size is set to 100 and we pre-train InstaAug for 10 epochs without augmentation. We train the model until convergence and the maximum epoch is set to 150.

**Contrastive training**    We directly apply InstaAug on the codebase[5] from Ermolov et al. (2021). Because of the characteristics of contrastive learning, we set the batch size to $512$. Same as the supervised case, we use SGD optimizer to train the augmentation module $\phi$. Differently, we use Adam optimizer (Kingma & Ba, 2015) (with learning rate $1e-3$ and weight decay $1e-6$) to train the base model. We train each model for $500$ epochs and decrease the learning rate by a factor of $0.8$ at step $450$ and $475$.

### D.2 Color jittering on textures

**Training.**    We use PreActResNet-18 ($width = 1$) on texture recognition task on RawFooT and train it with SGD optimizer. The learning rate is $0.02$ (with momentum $0.9$ and weight decay $1e-4$)

---

[4]https://github.com/alexrame/mixmo-pytorch.git, under Apache License v2.0.
[5]https://github.com/htdt/self-supervised.git, under Apache License v2.0.

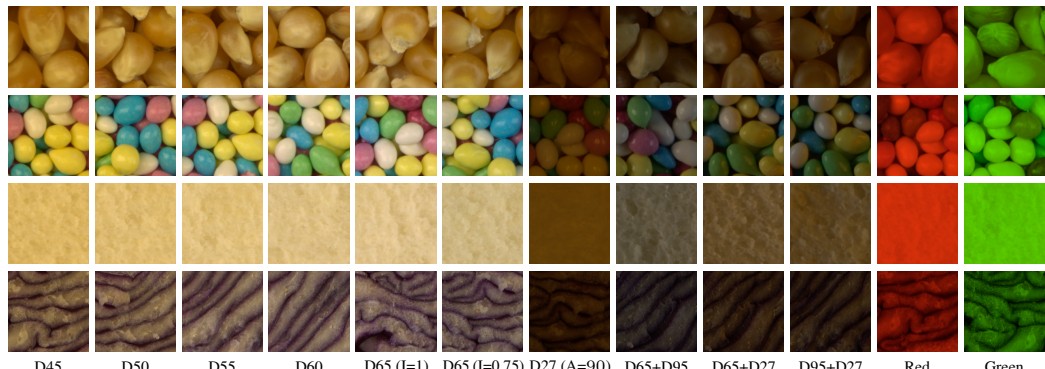

D45  D50  D55  D60  D65 (I=1)  D65 (I=0.75)  D27 (A=90)  D65+D95  D65+D27  D95+D27  Red  Green

Figure D.1: Examples of RawFooT data. Each row contains images in the same class (corn, candies, floor, red cabbage) under different lighting conditions. The left and right half of lighting conditions are in the easy and hard group, respectively.

for the classifier and $1e-5$ for the augmentation module $\phi$. We train each model for 50 epochs and learning rate schedulers are not necessary in this task.

**Random augmentation baseline.** We sweep over the variation range on each channel to find the best hyperparameters for the random augmentation baseline. For hue (h-jittering), we sweep between $[0, 0.5]$ with stride 0.1, and for saturation (s-jittering) as well as brightness value (v-jittering), we sweep between $[0, 1.0]$ with stride 0.2, which yields 216 different settings in total. The best accuracy shown in Table 3 is achieved where h,s,v= $0.0, 0.2, 0.8$.

**In-distribution vs. out-of-distribution generalization.** To further investigate the effect of each augmentation method, we additionally split the 46 test sets into two equally-sized groups. The first group contains lighting conditions similar to D45, such as daylight with different temperatures, for which the

Table D.1: Splitting of Lighting conditions.

| Group | Lighting id |
|-------|-------------|
| Easy (1) | 1-4,10,14-31 |
| Hard (2) | 5-9, 11-13, 32-46 |

vanilla model without augmentation trained on D45 has high test accuracy. The second group contains lighting conditions that are dramatically different from D45, for example, pure red light, which are more difficult for the vanilla method. Then the average accuracy on the first group can be regarded as a measure of in-distribution generalization, while the accuracy on the second group reflects out-of-distribution generalization.

## D.3 TIME COMPLEXITY

We notice that InstaAug on color jittering has a similar training speed (0.37s/iter) as random augmentation (0.40s/iter) on a single Nvidia 1080Ti GPU, though it takes more epochs (about 40) compared with random augmentation, which usually converges after 25 epochs. We also find the speed for evaluation is very fast even with test time augmentation (sample number =10), which is about 0.004s/sample. However, the training speed of InstaAug on cropping (0.25s/iter) is slower than random augmentation (0.15s/iter) due to optimization issues on the more complex parameterization method. Training InstaAug alone takes a similar amount of time for each epoch compared with joint training, but it requires fewer epochs (less than 30) to converge and we can cache the outputs of the classifier for faster training. The evaluation speed is 0.011s/sample when sample number is set to 50 for test-time augmentation.

## APPENDIX E   ADDITIONAL RESULTS AND DISCUSSION

### E.1 RAWFOOT

Figure E.1 shows some examples of learned color jittering. Though it's not easy to fully understand them, we can still find some patterns. For example, InstaAug tends to increase the brightness of

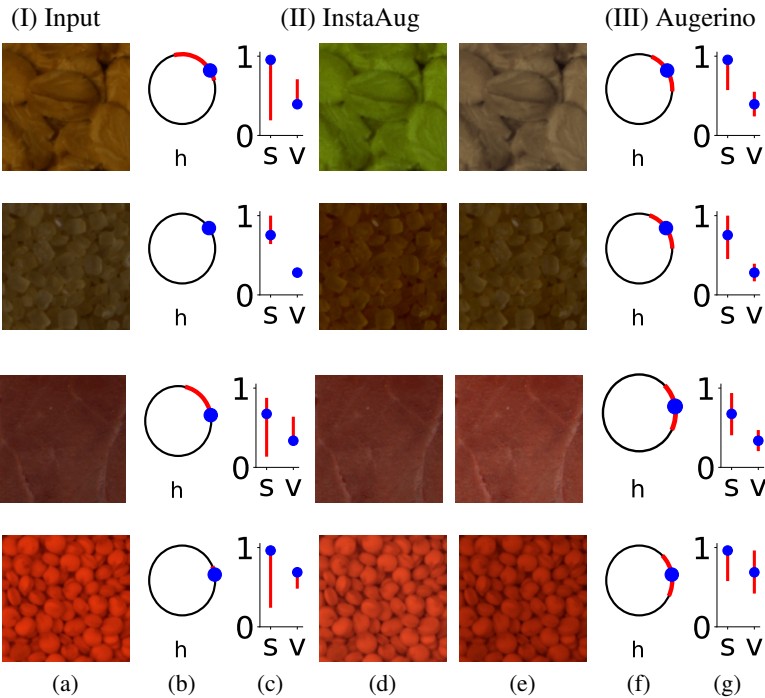

(I) Input      (II) InstaAug      (III) Augerino

(a)   (b)   (c)   (d)   (e)   (f)   (g)

Figure E.1: Examples of learned color jittering. (a) Original image; (b, f) Average hue (H) of original image (blue dot) and learned hue jittering (red arc) for InstaAug and Augerino; (c,g) learned saturation (S) and brightness value (V) of original image (blue dot) and learned hue jittering (red line segment) for InstaAug and Augerino; (d,e) examples of images transformed by InstaAug.

darker images (row 1 and 3) and decrease the brightness of brighter images (row 4). Also InstaAug is more likely to change saturation compared with hue and brightness, which is consistent with the common belief that saturation contains less information than hue and brightness.

InstaAug's behavior is quite different on different samples. It even decides not to augment the H and V channels of the image in the second row. In comparison, Augerino adds or multiplies noise to each channel with the same distribution across all samples, which is harmful in many cases. For example, the input image in the last row is already very bright. but Augerino allows further increasing its brightness. Then brightness values of many pixels will be capped at 1.0, which leads to loss of information.

### E.2 HYPERPARAMETER ABLATION

The two hyperparameters of InstaAug are $H_{min}$ and $H_{max}$, which reflect human preference on augmentation diversity. To investigate how $H_{min}$ and $H_{max}$ influence model performance and provide a guide on how to choose them, we perform an ablation study for the experiment of Section 5.2, wherein we sweep over possible intervals of length 0.5 and 1.0. From Table E.1, we find that the best accuracy is achieved when $[H_{min}, H_{max}]$ is set to $[3, 3.5]$, while any sub-interval of $[2, 4]$ produces significantly better results compared with random augmentation.

Table E.1: Model performance with different choice of $H_{min}$ and $H_{max}$ on supervised cropping.

| $H_{min}$ | $H_{max}$ | Accuracy (%) |
|---|---|---|
| 0.0 | 0.5 | 52.12 |
| 0.5 | 1.0 | 61.28 |
| 1.0 | 1.5 | 62.91 |
| 1.5 | 2.0 | 64.39 |
| 2.0 | 2.5 | 65.04 |
| 2.5 | 3.0 | 65.05 |
| 3.0 | 3.5 | **66.03** |
| 3.5 | 4.5 | 65.60 |
| 4.0 | 4.5 | 64.35 |
| 4.5 | 5.0 | 64.17 |
| 0.0 | 1.0 | 51.78 |
| 1.0 | 2.0 | 63.96 |
| 2.0 | 3.0 | 65.25 |
| 3.0 | 4.0 | **65.78** |
| 4.0 | 5.0 | 64.23 |

### E.3 WHY IS THE RANDOM AUGMENTATION BASELINE SO STRONG?

It is perhaps initially surprising that the Random Augmentation baseline in 5.2 is so strong compared to the other global augmentation schemes. In short, this occurs because the extensive hyperparameter

sweep used for it turns out to be a more effective tuning mechanism than directly training global parameters simultaneously to the model. To be more precise, for any *global* cropping scheme (which includes random crop, Augerino, and InstaAug without input), there is little to be gained from using a non-uniform distribution on the position of the crops. As such, the only thing that can be usefully learned is the distribution on the *size* of the crops themselves. For the random crop baseline, we do an exhaustive sweep to establish the best distribution on crop sizes, meaning that this baseline represents a near-optimal global cropping augmentation. By comparison, InstaAug (without input) must still learn the optimal cropping size distribution during training, and the results suggest that it does not always manage to do this perfectly, tending to prefer under-diverse transformations. This is perhaps not surprising, as it does not have access to a validation set, unlike the hyperparameter sweep implicitly being deployed for the random crop baseline. The problem is seen even more starkly for Augerino, where the lack of LRP causes training to become stuck in highly sub-optimal local optima that yield very little transformation diversity.

