# OpenReview forum: "Instance-Specific Augmentation: Capturing Local Invariances"
_ICLR.cc/2023/Conference — Submitted to ICLR 2023_

### Official Review · Reviewer_4op9 · 2022-10-24

**Confidence:** 3
**Correctness:** 3
**Technical Novelty And Significance:** 2
**Empirical Novelty And Significance:** 3
**Recommendation:** 6

**Clarity, Quality, Novelty And Reproducibility:**

Writing is clear and well-written in general, except for the comparison with closely related works. Parametrizing augmentations is not novel, but the specific way of location-related parametrization seems novel. I believe this work is reproducible with the provided code.

**Details Of Ethics Concerns:**

Nothing special.

**Strength And Weaknesses:**

Strengths

+ The task of capturing local invariances by learning data-specific augmentation is interesting.

+ The proposed method is simple but looks effective; by looking at the description, simply parameterizing data augmentations and optimizing its entropy is enough.

Weaknesses

- It seems prior works are not properly discussed. Based on the description in Section 5.2, the uniform parameterization is the same as Augerino, which is proposed by [Benton et al.]. Also, I believe test-time augmentation is also used in [Benton et al.]. Indeed, the performance of Augerino and the proposed method without LRP looks almost the same by looking at Table 1. Please do side-by-side comparison with closely related works like [Benton et al.].

- The first intuitive example in Figure 1(a) is digit recognition, but it is not appeared in the experimental results. I wonder if the range of rotation for 6 and 9 is indeed trained to be narrow, such that they are distinguishable.

- Similar to supervised learning, authors could also compare the proposed method with mixup-variants, which also have shown to be effective in many prior works, such as [Lee et al.] and [Shen et al.]. Also, I believe mixup-variants become a default choice to boost the performance of both supervised and self-supervised learning, so I wonder if the proposed method can be combined with them.

[Lee et al.] i-Mix: A Domain-Agnostic Strategy for Contrastive Representation Learning. In ICLR, 2021.

[Shen et al.] Un-Mix: Rethinking Image Mixtures for Unsupervised Visual Representation Learning. In AAAI, 2022.

**post-rebuttal**

The authors addressed my concerns mostly well, but an exception is the MNIST example in the motivative figure: I think their demonstration is indeed not proper and should be either completely removed or thoroughly discussed with experimental results to avoid any confusion.

Other reviewers found some critical points I was not initially aware of. For example, regarding AdaAug/MetaAug, if they indeed share the same spirit with the proposed method, then I think it is worth making some comparison (maybe by adapting the proposed method to the prior works' setting).

All in all, I do not change my initial rating, as I think other reviewers' concerns seem to be valid.

**Summary Of The Paper:**

This paper proposes an additional module that performs data-specific augmentation. Specifically, location-related parametrization taking account of spatial location has been proposed. Experimental results show the effectiveness of the proposed method on image datasets in supervised and self-supervised representation learning tasks.

**Summary Of The Review:**

I am mostly happy with this paper, except for the unclear comparison with prior works. Please answer my concerns above.

---

> ### Author Response · Authors · 2022-11-15
> **Reply to Reviewer 4op9**
>
> Thanks for your kind review and useful suggestions!
>
> &nbsp;
>
> > It seems prior works are not properly discussed. Based on the description in Section 5.2, the uniform parameterization is the same as Augerino, which is proposed by [Benton et al.]. Also, I believe test-time augmentation is also used in [Benton et al.]. Indeed, the performance of Augerino and the proposed method without LRP looks almost the same by looking at Table 1. Please do side-by-side comparison with closely related works like [Benton et al.].
>
> Thank you for the suggestion. We would like to highlight here that we already have a direct exposition of the differences with Augerino in Appendix B (which was Appendix A in the original submission).  However, we appreciate that we could have pointed to this more clearly in the main body of the paper.  We have made updates to correct this, and also edited Section 3.3 to further highlight the links and differences of the uniform parameterization to that of Augerino, noting that they are actually not the same: the limits of our uniform distribution depend on the input for InstaAug but not Augerino, while we drop the, potentially problematic, symmetry assumption Augerino makes.  You are correct that [Benton et. al.] also uses test-time augmentation and we have made this clearer.
>
>
> &nbsp;
>
> >  The first intuitive example in Figure 1(a) is digit recognition, but it is not appeared in the experimental results. I wonder if the range of rotation for 6 and 9 is indeed trained to be narrow, such that they are distinguishable.
>
> That is a great question. The reason why we use digits only as an illustrative example is we find that, for handwritten datasets like MNIST, the '6' digits are typically quite different from upside-down '9s'. Consequently, it is not difficult even for a simple fully connected neural network to distinguish between them, which makes it impossible to learn a boundary between them because they are actually not in the same 'orbit'.  The same concept though is demonstrated in Figure 5 for the cleaner 'Mario and Iggy' dataset, where we see that InstaAug does extremely well at keeping the classes distinguishable, while still keeping the rotation ranges as large as possible, typically extending to very near the true decision boundary.
>
>
> &nbsp;
>
> > Similar to supervised learning, authors could also compare the proposed method with mixup-variants, which also have shown to be effective in many prior works, such as [Lee et al.] and [Shen et al.]. Also, I believe mixup-variants become a default choice to boost the performance of both supervised and self-supervised learning, so I wonder if the proposed method can be combined with them.
>
> - This is an excellent suggestion: we agree that comparing to more mixup-variants would be helpful, noting they can also be regarded as data augmentation methods. Here we note first that we have already shown in Table 1 that InstaAug noticeably outperforms the mixup-variants MixMo, CutMix, and Mixup for supervised learning on Tiny-Imagenet.
>
> - To also provide a comparison for self-supervised learning, as per your suggestion, we have added Un-Mix as an additional baseline in Table 5, finding that it performs noticeably worse than InstaAug (49.58% compared to 55.05%).
>
> - The problem of whether InstaAug can be combined with mixup-variants is also a very interesting question. In principle, InstaAug can be applied directly when using these methods, either before or after mixing. This should be straightforward when using color jittering or rotation augmentations. However, since some mixup-variants (CutMix, Puzzle Mix) also crop original images to form training samples, it is less clear whether it will always be helpful to use InstaAug with cropping augmentations to crop these samples again. One idea here would be to use InstaAug to select the crops used for those mixup-variants, instead of additionally cropping samples before or after mixing.

---

> ### Author Response · Authors · 2022-12-07
> **Reply to 'post rebuttal' comments from reviewer 4op9**
>
> Thanks a lot for your approval and we will replace the MNIST figure with more proper examples.
>
> We'd love but find it difficult to empirically compare InstaAug with AdaAug/MetaAug. As reviewer 9vFe suggested, we need to modify AdaAug/MetaAug by using a new parameterization method and changing to InstaAug's loss function to compare with InstaAug in learning fine-grained augmentations. If we are otherwise comparing in their settings as you suggested, we would need to limit the power of InstaAug to learn low-level properties of each augmentation, (because AdaAug/MetaAug cannot learn them,) which would also be unfair.
>
> Though AdaAug/MetaAug might look similar to InstaAug in motivation, they work on completely different problems and it's not easy to compare them on either of the problems.

---

### Official Review · Reviewer_9vFe · 2022-10-25

**Confidence:** 4
**Correctness:** 3
**Technical Novelty And Significance:** 3
**Empirical Novelty And Significance:** 3
**Recommendation:** 5

**Clarity, Quality, Novelty And Reproducibility:**

Clarity: Some technical details are missing, but it is clear otherwise.
Code is not released although they provided link to a codebase that they started from.

**Strength And Weaknesses:**

Strengths.
1. Need for instance specific augmentation is well-motivated. I particularly like Section 5.1 and Figure 5.
2. Significant empirical gains in Table 1 and 2.


Weakness/questions.
1. The proposed method is simple, which is nice. However, in (2b), I do not see what pushes the per-instance transformation entropy to be more than H_min. In other words, what part of the loss encourages diversity in the distribution of transformations?
2. Comparison with per-label (instead of per-instance) augmentations should also be presented.
3. In Table 1, InstaAug (without input) but with LRP is worse than global random crop, which is unexpected. I expected them to be at least comparable, why do we see a difference?
4. Section 5.3 does not describe in detail how \phi is re-learned and on what data.  Technical details on how \phi is parameterized is also missing. Section 3.3 is also not very clear.
5. With regard to test-time augmentations of Section 5.3, the improvement in performance with InstaAug is impressive. However, InstaAug can only learn augmentations that the trained model is already robust toward, I do not see the source of improvement in Table 2. Could you explain/comment?
6. (Minor) Figure 7 contains some legends but not their plots making it look incomplete.


**Summary Of The Paper:**

Data augmentations with common input transformations is a commonplace in ML training, and is known to improve performance. This work argues that example-agnostic common transformations are too restrictive and that example specific augmentations can allow much wider transformation. The proposed methods of learning instance specific augmentations is shown to improve performance when training on Tiny-Imagenet, test-time augmentations and for contrastive learning.

**Summary Of The Review:**

I am not yet fully convinced that the paper is foolproof. The motivation and problem setup is clear, but their method and technical details are not. I request the authors to address my concerns listed above and I will make a more decisive evaluation after looking at authors’ responses and based on other reviews.

--------
**Edit after response.**
I thank the authors for patiently addressing all the question. Most of my question/confusion have now been resolved. I thank the authors again for adding the comparison with per-label augmentations. After reading other reviews and author comments, I have the following outstanding concerns.

* Reviewer eZc6 pointed to several related work. I agree with eZc6 that the paper should have elaborated more on their existence and comparison. I find the reasons the authors provide for not comparing with MetaAug/AdaAug unconvincing for the following reasons: (a) "MetaAug/AdaAug do not search over parameterisation of augmentations", which could perhaps be fixed simply by extending the search space to include augmentations with different parameterisations. (b) "they require validation dataset", given that train loss is a good proxy for InstaAug, it could also be a good proxy for MetaAug/AdaAug. Why not simply replace their need for validation dataset with train dataset? I agree that it is not trivial to use them for comparisons on test-time augmentations though. I did not go over the related work before and got the impression (after reading the draft) that instance specific augmentations is novel when it is not. The paper thus may need to be restructured to carefully communicate their contributions.
* Reviewer 4op9 raised an important point about evaluation on the running examples of digit rotations. The authors' response to this question is concerning. If instance-specific augmentations are not useful in this simple example, I do not understand how it is helping for other (complex/real-world) datasets. In that regard, the paper should present a more faithful explanation/intuition of their method.

Existing work on learning effective augmentation stratgies (including instance-specific augmentations) all use policy search based methods and therefore InstaAug is more desirable since it learns the augmentation policy end-to-end. LRP and parameterising of augmentations are all interesting contributions, which make InstaAug readily available for examples beyond training data such as in test-time augmentations. Although I am excited and recognise these merits of InstaAug, I am discourcaged by problems in exposition and comparison. The paper is an interesting contribution (especially given their performance gains) at some point, but the current draft needs further work. To make the acceptance decision more easier, I am now lowering my earlier score.

---

> ### Author Response · Authors · 2022-11-15
> **Reply to Reviewer 9vFe (Part 2)**
>
> >5. With regard to test-time augmentations of Section 5.3, the improvement in performance with InstaAug is impressive. However, InstaAug can only learn augmentations that the trained model is already robust toward, I do not see the source of improvement in Table 2. Could you explain/comment?
>
> Good question! In short, it can inductively bias the predictions towards those which generalize well.  In particular, it can help alleviate any overfitting in the original model to input variations our predictions should be invariant to.  For example, we experimentally find that the original model sometimes misclassifies the original input, but correctly classifies cropped patches which are focused on the main object in the image.  We note here that we are not the first to show that the test-time augmentation can provide improvements (see e.g. https://arxiv.org/abs/2005.00178, https://arxiv.org/abs/2010.09515), even though we find that current baselines approaches are actively harmful to this problem.
>
>
> &nbsp;
>
> >6. (Minor) Figure 7 contains some legends but not their plots making it look incomplete.
>
> We believe this is actually a misunderstanding: plots for all four methods are present, but the InstaAug and No aug methods only represent single points (top right and bottom right respectively) that are easily missed.  The many points for Random aug represent the many different hyperparameter configurations tested, with a similar search not required for InstaAug and No aug.
>
> &nbsp;
>
> >* Code is not released although they provided link to a codebase that they started from.
>
> Our code is actually available in the supplementary zip file and we plan to release it publicly upon publication. For convenience, we have also now set up an anonymous github repo that can be accessed here: <https://github.com/lkwlkwlkwlkewlkr/InstaAug>.

---

> ### Author Response · Authors · 2022-11-15
> **Reply to Reviewer 9vFe (Part 1)**
>
> Thanks a lot for your detailed and helpful review!
>
>
> &nbsp;
>
> >1. The proposed method is simple, which is nice. However, in (2b), I do not see what pushes the per-instance transformation entropy to be more than H_min. In other words, what part of the loss encourages diversity in the distribution of transformations?
>
> We think there might be a minor misunderstanding here as we are not generally trying to push the entropy to be _more_ than $H_{\mathrm{min}}$, we only care about making sure the entropy is _at least_ $H_{\mathrm{min}}$. In practice, we are explicitly driving towards higher entropies through the $\lambda \mathbb{E}[\mathbb{H}[p(\tau;\phi(x))]]$ term in the Lagrangian used for the optimization (as per the last paragraph of Section 3.2).
>
> We could have been clearer though and have made updates.
>
> &nbsp;
>
> >2. Comparison with per-label (instead of per-instance) augmentations should also be presented.
>
> Thanks for this great suggestion! To provide such comparisons, we built a label-specific variant of InstaAug, which takes in sample labels rather than images as input. We added this as a new baseline in Table 1.  Interestingly, we find that the performance of this approach is not only much weaker than InstaAug, but actually worse than some of the global augmentation baselines as well (including InstaAug (without input)).  We believe this is because of difficulties in performing appropriate test-time augmentations for such an approach, with the class labels no longer available such that we have to resort to a distinct augmentation scheme than that used in training.
>
>
> &nbsp;
>
> >3. In Table 1, InstaAug (without input) but with LRP is worse than global random crop, which is unexpected. I expected them to be at least comparable, why do we see a difference?
> This is a great point and we agree it could have been made clearer. In short, the gap is because the extensive hyperparameter sweep used for the random crop baseline turns out to be a more effective tuning mechanism than directly training global parameters simultaneously to the model.
>
> To be more precise, for any _global_ cropping scheme (which includes random crop, Augerino and InstaAug without input), there is little to be gained from using a non-uniform distribution on the position of the crops.  As such, the only thing that can be usefully learned is the distribution on the _size_ of the crops themselves.  For the random crop baseline, we do an exhaustive sweep to establish the best distribution on crop sizes, meaning that this baseline represents a near-optimal global cropping augmentation.  By comparison, InstaAug (without input) must still learn the optimal cropping size distribution during training, and the results suggest that it does not always manage to do this perfectly, tending to prefer under-diverse transformations.  This is perhaps not surprising, as it does not have access to a validation set, unlike the hyperparameter sweep implicitly being deployed for the random crop baseline.  The problem is seen even more starkly for Augerino, where the lack of LRP causes training to become stuck in highly sub-optimal local optima that yield very little transformation diversity.
>
> We have made edits to make this clearer, including adding the above discussion to the paper itself in Appendix E.3.
>
> &nbsp;
>
> >4. Section 5.3 does not describe in detail how \phi is re-learned and on what data. Technical details on how \phi is parameterized is also missing. Section 3.3 is also not very clear.
> Thank you for highlighting these problems, we agree that these sections were less clearly written than the rest of the paper and have duly updated them.
>
> In Section 5.3, $\phi$ is learned in exactly the same way as elsewhere: it uses the training procedure of Section 3.2, the (Imagenet) training data, and the location-related parameterization described in Section 3.3.  The only thing that is different is that $f$ is no longer being learned simultaneously, but is instead fixed to a pre-trained ResNet-50 (which did not itself use an invariance module during training).  We then directly deploy the $\phi$ learned using this ResNet-50 for test-time augmentation on the two other pre-trained classifiers (ResNet18 and XCiT) _without_ any re-learning, demonstrating its effective generalization.
>
> We have significantly updated the writing in Section 3.3 to provide a better explanation of the parameterizations. We hope you find it clearer than the original.

---

> ### Author Response · Authors · 2022-12-02
> **Reply to 'Edit after response' from Reviewer 9vFe**
>
> Thanks for your further comments!
>
> - You mentioned that we could modify MetaAug/AdaAug to learn each augmentation like InstaAug by (a) extending their search space with different parameterisations and (b) replacing meta loss on validation set with training loss. Though we believe the method could work, we don’t think the modifications are trivial. Firstly, there isn’t a readily available parameterization method to finely control augmentations such as cropping, which is why we designed LRP. Secondly, It’s not feasible to directly train the augmentation module to decrease training loss, as this would result in augmentations with very low diversity- this is precisely why we introduced the entropy term in InstaAug. Overall, your proposal is not a simple application of MetaAug/AdaAug, and the issues it highlights are direct motivations for the method we actually propose. We consider it unreasonable to criticize the paper for lack of comparison with a non-existent method.
>
> - Though it’s impossible to learn the partial invariance for some ‘9’s and ‘6’s, we find that InstaAug can still correctly predict the partial invariance of other figures of ‘9’ and ‘6’, which is helpful for the classifier compared with random rotation. The reason we didn’t show results and examples from MNIST is 1) the baseline accuracy is too high (99%+) to make significant improvements and 2) we don’t want to mislead our readers to think that InstaAug can correctly learn the rotation invariance for every ‘6’ and ‘9’, unlike the Mario-Iggy case shown in Section 5.1.

---

### Official Review · Reviewer_uSWY · 2022-10-27

**Confidence:** 4
**Clarity, Quality, Novelty And Reproducibility:** The paper is original as far as I kno…
**Correctness:** 4
**Technical Novelty And Significance:** 3
**Empirical Novelty And Significance:** 3
**Recommendation:** 6

**Strength And Weaknesses:**

Strength:
1. The algorithm is simple and effective.
2. It applies to a wide range of scenarios, including supervised learning and contrastive learning.

Weakness:
1. The two principles suggested at the beginning of Section 2, while intuitively correct, are not theoretically justified in the paper. In addition to empirical successes, is there a reason why these two principles lead to better representation? The authors may need to explain them from an information-theoretical perspective (since the loss function includes entropy) or from a learning-theoretical perspective.
2. The proposed method introduces too many hyperparameters. For example, the algorithm introduces lower/upper limits of the entropy and the regularizer in the loss function. Furthermore, the algorithm requires a set of hyperparameters for each type of augmentation. Therefore, it is not easy to adopt the algorithm to a new scenario (without a tedious hyperparameters search).
3. The current method only works for vision models since the augmentation types are mostly vision-specific. However, the proposed method should also work in other domains, particularly natural language processing. It will make the technique more general/impactful by demonstrating its capabilities other than computer vision.

**Summary Of The Paper:**

The paper proposes an algorithm to learn instance-specific augmentation for images, which substantially improves computer vision models over a wide range of scenarios.

**Summary Of The Review:**

The paper proposed a simple and effective instance-specific algorithm that generally improves the performance of computer vision models. However, there are still some limitations, as suggested in Strength And Weaknesses.

---

> ### Author Response · Authors · 2022-11-15
> **Reply to Reviewer uSWY**
>
> Thank you for your helpful review! We have updated the submission file as per your suggestions with a new theoretical analysis section as well as empirical results comparing our methods with per-class augmentations.
>
> &nbsp;
>
> >1. The two principles suggested at the beginning of Section 2, while intuitively correct, are not theoretically justified in the paper. In addition to empirical successes, is there a reason why these two principles lead to better representation? The authors may need to explain them from an information-theoretical perspective (since the loss function includes entropy) or from a learning-theoretical perspective.
>
> That’s a great suggestion! We have actually managed to carry out exactly such an analysis: we have added a comprehensive new theoretical Section in Appendix A, which analyzes the problem from a learning perspective and shows that these principles naturally emerge from the generalization error that results from using the invariance module during training. In short, this new analysis shows that the generalization performance of a classifier trained with data augmentation depends on a combination of (A) how well the transformations preserve the conditional label distribution and (B) the discrepancy between the real data distribution (which is unknown) and the augmented empirical data distribution. The former exactly corresponds to the principle of preserving the label information, while the latter is shown to be directly linked to the principle of ensuring transformation diversity.  The new analysis also provides explicit justification for some of the specific algorithmic constructions in the InstaAug setup.
>
>
> &nbsp;
>
> >2. The proposed method introduces too many hyperparameters. For example, the algorithm introduces lower/upper limits of the entropy and the regularizer in the loss function. Furthermore, the algorithm requires a set of hyperparameters for each type of augmentation. Therefore, it is not easy to adopt the algorithm to a new scenario (without a tedious hyperparameters search).
>
> Though InstaAug does certainly contain hyperparameters, we do not feel that it is true that it has a large number of them or that tuning them is any more difficult than with other augmentation methods (all of which have important hyperparameters of their own).  Firstly, we only have two hyperparameters in $H_{\mathrm{min}}$ and $H_{\mathrm{max}}$, noting that $\lambda$ is not a fixed hyper-parameter, but a Lagrange multiplier that is automatically adaptively adjusted during training. Itself does not require any tuning between problems.  Secondly, these are both intuitive scalar values that are easy to train and which the method exhibits a good degree of robustness to within a sensible range.
>
> To show how $H_{\mathrm{min}}$ and $H_{\mathrm{max}}$ influence model performance and how to tune them, we have added a new ablation study in Appendix E.2. We find in supervised cropping, InstaAug easily beats random augmentation whenever $[H_{\mathrm{min}}, H_{\mathrm{max}}] \subset [2, 4]$, which is not difficult to find, given the entropy is only ever able to be in the range $\in [0, 6]$ for most datasets and parameterization methods.
> It is usually not difficult to choose $H_{\mathrm{min}}$ and $H_{\mathrm{max}}$ for other augmentations based on prior knowledge. For the rotation experiment, because we expect each image to be rotated at least $\pi$ degree, and at most $2\pi$ to avoid generating repeated samples, we set $[H_{\mathrm{min}}, H_{\mathrm{max}}] = [\pi, 2\pi]$. For color jittering, because we find image quality is usually more sensitive to brightness than saturation, we set $[H_{\mathrm{min}}, H_{\mathrm{max}}]  = [0.1, 0.6]$ for brightness, and $[H_{\mathrm{min}}, H_{\mathrm{max}}]  = [0.5, 1.5]$ for saturation.
>
>
> &nbsp;
>
> >3. The current method only works for vision models since the augmentation types are mostly vision-specific. However, the proposed method should also work in other domains, particularly natural language processing. It will make the technique more general/impactful by demonstrating its capabilities other than computer vision.
>
> We wholeheartedly agree that InstaAug can be applied to different modalities and tasks, and that it would be great to demonstrate this. Indeed, this should not require any major changes to the general method formulation.  Our focus on vision tasks was driven by the fact that a) it is currently the most common setting for augmentation methods with the most established approaches for transforming inputs, and b) limitations on time and space, with a wide variety of experimentation already undertaken tasks like NLP requiring noticeable extra work that is tangential to our actual contributions (e.g. reconfiguring baselines).  Nonetheless, we will endeavour to add results on a new data modality for the camera ready submission.

---

### Official Review · Reviewer_eZc6 · 2022-11-24

**Confidence:** 4
**Correctness:** 3
**Technical Novelty And Significance:** 2
**Empirical Novelty And Significance:** 2
**Recommendation:** 5

**Clarity, Quality, Novelty And Reproducibility:**

The paper is well-written and easy to follow. Some components are quite novel, but overall the idea of instance-adaptive augmentation is not new. Differences from related works are not well described in the draft, and experimental comparisons are too limited. The authors submitted their code, so I believe this work is reproducible.



**Strength And Weaknesses:**

(+) The paper is well-written and the proposed method is simple and straightforward.

(+) The location-related parameterization (LRP) for cropping seems quite novel to me.

(-) It seems the flow of the draft needs to be substantially modified. Most parts of the draft describe the necessity of instance-wise augmentation in comparison to global augmentations. As the idea of instance-wise (or class-wise) augmentation is not new [1,2], the differences and empirical comparisons with these works should be more intensively addressed in the paper. Only brief and limited descriptions can be
found in the related works section. The authors claim that these prior works focus on choosing ‘which type of transformation to apply’ while InstaAug keeps the type of transformation fixed and learns instance-specific parameters. However, [1] searchs not only the type of transformation but also the magnitude, and in this regard, the advantage of ‘keeping the transformation fixed’ is unclear given that
a set of hyperparameters have to be found for each transformation.

(-) Some of the motivations for the proposed approach are not convincing. The authors explain the limitations of global augmentation approaches using Figure 1. In Figure 1 (b), it seems the global augmentation could rather boost the generalization performance since the model has to exploit other features (e.g. texture, shape) for correct prediction without simply relying on color. I believe this is why
Cutmix and Cutout have been successful as it prevents the model from predicting only highly discriminative features. Meanwhile, the proposed method might fortify the inductive bias of the classifier as the $\phi$ is trained to simply reduce the classification error of the classifier. It would be great if the authors can provide evidence/analysis to further strengthen their claim.

(-) Section 3.2 and Figure 3 are weak to support the authors’ claim. Of course, extreme global transformations will generate incorrect samples (‘red’ region in Figure 3). But it seems the proposed method also suffers from a similar issue and probably this is the reason why the authors introduce the upper bound $H_{max}$ in their constraints. Wouldn't this problem be solved by relaxing the type and
intensity of transformation in global augmentation as well? (why should Figure 3 be an illustration rather than the visualizations of the actual decision boundary of Global Augmentation and InstaAug?)

(-) This work aims to propose an augmentation that can conduct information-preserving transformations while keeping a sufficient diversity of transformations. Inspired by goals similar to this work, saliency-based augmentations [3, 4, 5, 6] are recently proposed for information-preserving augmentations. [5, 6] also consider the diversity of augmented data. Although previous related works did not compare with these works, for this work, empirical comparisons with these saliency augmentations are required given the highly similar goals (at least combining InstaAug with these saliency augmentations). Comparing the method against baselines such as Cutmix and Mixup is
insufficient.

---
### Questions
- Does this method work well for data-deficient conditions? Since data augmentation is most needed when data is scarce, I wonder if the invariance module can learn the proper local invariance even when there is insufficient data.

- Gradient-based saliency [5,7] (or other saliency methods) can detect instance-specific regions without relying on predefined sets. What are the advantages of LRP over these?

- Although this fact doesn't affect my score, is there a reason to experiment with only one dataset per augmentation type? (e.g. TinyImageNet can be used also for testing color jitter). It would be good to experiment with commonly used datasets such as CIFAR and SVHN.

- (minor) The tuning for $H_{min}$ and $H_{max}$ is necessary when practitioners deploy this method to their own data modality or augmentations. It would be nice if the insights of determining $H$ are explained in the main paper

[1] Cheung, Tsz-Him, and Dit-Yan Yeung. "AdaAug: Learning Class-and Instance-adaptive Data
Augmentation Policies." International Conference on Learning Representations. 2021.

[2] Zhou, Fengwei, et al. "Metaaugment: Sample-aware data augmentation policy
learning." Proceedings of the AAAI Conference on Artificial Intelligence. Vol. 35. No. 12. 2021.

[3] Kim, Jang-Hyun, Wonho Choo, and Hyun Oh Song. "Puzzle mix: Exploiting saliency and local
statistics for optimal mixup." International Conference on Machine Learning. PMLR, 2020.

[4] Uddin, A. F. M., et al. "Saliencymix: A saliency guided data augmentation strategy for better
regularization." arXiv preprint arXiv:2006.01791 (2020).

[5] Park, Joonhyung, et al. "Saliency grafting: Innocuous attribution-guided mixup with calibrated label
mixing." Proceedings of the AAAI Conference on Artificial Intelligence. Vol. 36. No. 7. 2022.

[6] Kim, JangHyun, et al. "Co-Mixup: Saliency Guided Joint Mixup with Supermodular
Diversity." International Conference on Learning Representations. 2020.

[7] Simonyan, Karen, Andrea Vedaldi, and Andrew Zisserman. "Deep inside convolutional networks:
Visualising image classification models and saliency maps." arXiv preprint arXiv:1312.6034 (2013)

**Summary Of The Paper:**

This paper elucidates the importance of instance-dependent transformation in augmenting data and proposes InstaAug that learns the input-dependent transformation which could preserve the information of original input while maintaining sufficient transformation diversity. The invariance module $\phi$ is jointly trained with the classifier $f$, which can predict the parameters of transformation distribution based on a given input. The authors also propose a location-related parameterization technique to parameterize cropping augmentations.

**Summary Of The Review:**

Overall, I think this work needs further refinement for acceptance due to the concerns mentioned above (please see Strength And Weakness section). I understand that it is no longer possible for the authors to revise their draft, but I would like to see the authors' response within the remaining period.

---

> ### Author Response · Authors · 2022-11-28
> **Reply to Reviewer eZc6 (Part 4)**
>
> >* Although this fact doesn't affect my score, is there a reason to experiment with only one dataset per augmentation type? (e.g. TinyImageNet can be used also for testing color jitter). It would be good to experiment with commonly used datasets such as CIFAR and SVHN.
>
> This was simply down to restrictions on time, computation, and space in the paper itself.  While we would love to add more such testing, there are inevitably limitations on what can be done for a single paper (especially as our computational resources are quite limited).
>
> &nbsp;
>
> >* The tuning for Hmin and Hmax is necessary when practitioners deploy this method to their own data modality or augmentations. It would be nice if the insights of determining H are explained in the main paper.
>
> Thanks for this suggestion! We will gladly move some key points from appendix E.2 to the main paper.

---

> ### Author Response · Authors · 2022-11-28
> **Reply to Reviewer eZc6 (Part 3)**
>
> > Wouldn't this problem be solved by relaxing the type and intensity of transformation in global augmentation as well?
>
> This is an interesting question. Relaxing the type of augmentation could indeed sometimes help, but this relaxation can be equally applied to both global augmentations and InstaAug.  As such, it is an orthogonal design decision: we focus on learning the best way to perform each specific augmentation and show that being instance-specific provides substantial benefits in this regard.
>
> Relaxing the intensity of the transformation in global augmentation is something that is already explicitly investigated, as we perform extensive hyperparameter searches for our baselines to directly optimize this.
>
>
> &nbsp;
>
> > Why should Figure 3 be an illustration rather than the visualizations of the actual decision boundary of Global Augmentation and InstaAug?
>
> Thanks for this sensible suggestion. Figure 5 provides such a visualization from real experiments with the actual decision boundaries in the case of rotations. We felt it might be difficult for the reader to parse this plot before the InstaAug method itself was fully described, but are happy to bring this forward to replace Figure 3 if you think that would be useful.
>
>
> &nbsp;
>
> > Comparing the method against baselines such as Cutmix and Mixup is insufficient. Should compare with saliency-based augmentations [3, 4, 5, 6].
>
> Thanks for this great suggestion, we are very happy to add these comparisons and provide more discussion on saliency-based augmentation in the paper. Luckily, [5] and [6] use the same setup and ResNet-18 model as we do in Section 5.2, so we can immediately provide these comparisons using their reported numbers in the following table.
>
> &nbsp;
>
> | Method | Puzzle-Mix | Co-Mixup | Saliency grafting| InstaAug|
> | :--- | :---: | :---: | :---: | :---: |
> |Acc(%)   | $63.48^{[6]}$| $64.15^{[6]}$ | $64.84±0.12^{[5]}$ | __$66.02±0.18$__ |
>
>
> We will add these results to our paper.
>
> &nbsp;
>
> >* Does this method work well for data-deficient conditions?
>
> Good question! Yes, we certainly believe this to be the case and that the paper provides strong evidence to this effect. Firstly, Tiny-Imagenet was deliberately chosen as a common benchmark for the data-deficient setting.  Secondly, Table 4 provides a specific ablation for performance as the amount of data is reduced and provides strong evidence that InstaAug can work well in low-data regimes. In fact, _we find the gains over the baselines got larger as the amount of data was reduced_.  For example, when we are training on only $1$ lighting condition, with a single training sample for each class, InstaAug achieves 7.5% and 3.3% accuracy increase compared with no augmentation and random augmentation.
> We will make edits to the paper to emphasize this point more clearly.
>
> &nbsp;
>
> >* Gradient-based saliency [5,7] (or other saliency methods) can detect instance-specific regions without relying on predefined sets. What are the advantages of LRP over these?
>
>
> This is an interesting question. Unfortunately, though, because these saliency-based augmentation approaches work very differently from InstaAug, their underlying patch-selection methods cannot be used as an alternative to our LRP to parameterize a _learnable invariance module_.  They could, in principle, be used as an alternative mechanism for generating instance-specific croppings, but the lack of a learnable invariance module would mean such an approach falls outside the InstaAug framework. This hypothetical approach would also have a number of disadvantages, such as not allowing transferral of the learned module to test-time augmentation and different models (noting we find in Table 2 that InstaAug is able to learn invariances that generalize between models), lacking natural applicability to unsupervised settings, and potentially restricting the architectures that can be used for the downstream model.
>
> In addition to the gains from having an explicit invariance module, there are also some additional key advantages of using InstaAug over saliency-based augmentation methods in general:
> - The earlier empirical results show significant performance improvements from using InstaAug compared with saliency-based methods.
> - Saliency-based methods are specific to cropping and do not apply to more general augmentations.
> - Current saliency-based augmentation approaches cannot be used with pre-trained models or in unsupervised settings.

---

> ### Author Response · Authors · 2022-11-28
> **Reply to Reviewer eZc6 (Part 2)**
>
> >* Some of the motivations for the proposed approach are not convincing…  In Figure 1 (b), it seems the global augmentation could rather boost the generalization performance since the model has to exploit other features (e.g. texture, shape) for correct prediction without simply relying on color… the proposed method might fortify the inductive bias of the classifier… It would be great if the authors can provide evidence/analysis to further strengthen their claim.
>
> Though we certainly agree that global augmentation can have better generalization performance compared with no augmentation, we believe that we do already provide extensive evidence and analysis to show that it is not as effective as the instance-specific augmentations:
> - We provide extensive empirical evidence that InstaAug significantly outperforms global augmentation schemes: we did not find any global augmentation preferable to InstaAug in any of our experiments.
> - While we agree that using an overly flexible class of transformation distributions could lead to increased overfitting and worse generalization, this is exactly why we already discuss the need to use restricted transformation classes that reflect our desired inductive biases (cf Sections 1, Paragraph 3 and Section 2).  These are specifically set-up to influence how we would like the classifier to generalize, so should generally correct unwanted biases / overfitting in the classifier, not reinforce them.  Indeed, this is exactly the mechanism by which InstaAug can be helpful when only used for test-time augmentation. Critically, we again saw no empirical evidence of InstaAug having negative effects in this regard.
> - Our theoretical analysis provides a clear insight into why InstaAug will typically provide better generalization errors than global augmentation.  It shows that generalization performance is harmed when augmentations do not preserve the label information and/or fail to capture the full input distribution.  Global augmentations are limited in their ability to simultaneously meet these needs for real problems where the transformation distribution is inevitably restricted.  This is because it is not generally possible to properly cover the input space, without either varying the transformation between inputs or inducing changes in the labels (as is visually demonstrated in Figure 5).
> - Reducing the generalization error is only useful if the gains are bigger than any increases in the training error itself, with global augmentations generally leading to higher training errors in the first place.  For example, in Figure 1b removing the reliance on color will only be helpful if the reduction in overfitting is more significant than the loss of relevant information. If there are other significant features that distinguish between a lime and a lemon, InstaAug can also learn large hue jittering to increase transformation diversity. If not, InstaAug can avoid the increases in training error caused by excessive transformations, without harming the generalization performance as per the reasons above.
>
> We will make edits to the paper to further elucidate these points. Please do let us know if you think there is any further specific experimentation or analysis you think would be helpful to add to this evidence.
>
>
> &nbsp;
>
> >* Section 3.2 and Figure 3 are weak to support the authors’ claim. Of course, extreme global transformations will generate incorrect samples (‘red’ region in Figure 3). But it seems the proposed method also suffers from a similar issue and probably this is the reason why the authors introduce the upper bound Hmax in their constraints.
>
> We are sorry to hear you did not find this section more convincing and will make edits to make our points clearer.
> Because global augmentations are applied identically to all inputs, they typically do not need to be extreme to lead to _some_ ‘incorrect’ samples, which can harm model performance. Moreover, the aforementioned need to cover as much of the input space as possible means there is always in trade-off with the need to avoid such errors.
>
> Figure 3 explains why this trade-off is difficult to manage with global augmentations, while Figure 5 and 6 show how this is much better managed by InstaAug.  We would also again here like to point to the fact InstaAug outperforms (learnable or random) global augmentation in all of our experiments (see Tables 1-5).  Together these provide strong evidence that InstaAug does not suffer from the same issues, at least not to the same extent as global augmentation.
>
> $H_{\text{max}}$ is introduced for quite different reasons than avoiding errors in the final solution: it is only required for stability during the early stages of training and was never close to being an active constraint in the final trained models.

---

> ### Author Response · Authors · 2022-11-28
> **Reply to Reviewer eZc6 (Part 1)**
>
> Thanks for serving as an emergency reviewer for our paper. We very much appreciate your feedback, particularly when it was given at such short notice. We hope the following responses alleviate your concerns.
>
> &nbsp;
>
> > As the idea of instance-wise (or class-wise) augmentation is not new [1,2], the differences and empirical comparisons with these works should be more intensively addressed in the paper… [1] searches not only the type of transformation but also the magnitude.
>
> Thank you for this suggestion. While we will happily increase the level of discussion on these works, mention them earlier, and make it clearer that [1] allows for some control on the magnitude of the transformations, we would also like to highlight that neither of these papers propose methods to _learn_ instance-specific transformation distributions or local invariances themselves, as per the key aims of InstaAug.  Namely, [2] only reweights existing transformations without affecting the transformations themselves.  [1], meanwhile, only allows for limited control of the transformations and is mostly focused on class-conditional augmentations: it only claims to learn a “potentially instance-dependent augmentation policy”.  In particular, its policy is based on a linear mapping from the penultimate layer of the classification model to probabilities and magnitudes over a finite set of possible augmentations, so it does not have the fine-grained control to learn local invariances or the transformation distribution itself.  For example, they only ever learn to turn cropping on or off, while InstaAug learns a full invariance module that provides a distribution over patch sizes and positions for each input.
>
> Fair empirical comparison to these methods is, unfortunately, not generally feasible, as their requirements are quite different from those of InstaAug. Namely, they require a separate validation set, only consider augmentation during training, and cannot be applied in unsupervised settings.  As such, comparisons cannot be run at all for the experiments in Sections 5.3 and 6, because these are based on test-time augmentation and unsupervised augmentation respectively.  Meaningful comparison is also not possible in Section 5.2 (beyond those already made), as AdaAug/MetaAug do not make local adjustments to the crops that are made, so they will reduce to a global augmentation scheme like those already benchmarked against.  It might be possible to make some comparisons in Section 5.4, but the lack of a validation set here means that AdaAug/MetaAug will need to sacrifice some of the training data to provide it, inevitably harming their performance.
>
> &nbsp;
>
> > The advantage of ‘keeping the transformation fixed’ is unclear given that a set of hyperparameters have to be found for each transformation.
>
> We believe there may have been a misunderstanding here as we are not advocating for keeping the transformation fixed, just proposing an approach for how to learn the transformation distribution given a particular distribution class.  InstaAug is actually very much compatible with simultaneously using AdaAug or MetaAug to learn a policy for choosing between different types of augmentation: we simply need to introduce a separate invariance module for each of the different augmentation types.  This is another important reason why it is difficult to make meaningful empirical comparisons between InstaAug and these methods: it is intended as a complementary rather than competing method. We will make edits to make this clearer.

---

### Author Response · Authors · 2022-11-15
**General Response**

We thank all reviewers for their time, insightful comments, and helpful feedback. We are thrilled that all reviewers like the core idea of this work, and that our method was generally regarded to be simple and effective. We have updated the paper according to the concerns and suggestions raised. In addition to answering reviewers' questions individually, we also wanted to highlight the following key changes we’ve made to the paper:

* Added a substantial new theoretical section (Appendix A) that analyzes the key factors influencing the performance of data augmentation methods and rigorously elucidates the key motivations behind InstaAug’s construction.
* Added a per-class variant of InstaAug as a new baseline in the supervised cropping experiments (Section 5.2).
* Added a new mix-up baseline for the contrastive learning experiments (Section 6).
* Added an ablation study on the choice of hyper-parameters $H_{\mathrm{min}}$ and $H_{\mathrm{max}}$ (Appendix E.2).

---

### Decision · Program_Chairs · 2023-01-20

**Decision:**

Reject

**Justification For Why Not Higher Score:**

see 'summery of ac-reviewer meeting'

**Justification For Why Not Lower Score:**

n/a

**Metareview: Summary, Strengths And Weaknesses:**

This paper proposes a method called InstaAug that learns input-specific augmentation directly from training data. A total of 5 reviewers reviewed this paper; 2 reviewers suggested borderline accept and the remaining 2 suggested borderline reject. Most of the reviewers agreed that the paper is clearly well written and has the advantage of proposing a simple yet effective algorithm. However, despite the authors' rebuttal, this AC and the reviewers were not completely convinced of the following two issues, and in the additional ac-reviewer meeting, we also pointed out them as critical problems of this paper:

- lack of technical novelty : considering two previous works i) Augerino learning augmentation strategy only through training objective/data without valuation set and ii) papers like AdaAug (mentioned by Reviewer eZc6) learning the augmentation method instance-wisely. The proposed method can be understood as the simple combination of two approaches. In particular, this paper seems to overemphasize the instance-wise augmentation learning as the original contribution of this paper.

- lack of strong experimental evidence : i) the effect of  instance specific aug is not clear if we look at table 1 (as mentioned by reviewer 4op9). LRP may be novel but it alone is not good enough for the acceptance. ii) missing lots of recent baselines for augmentations. The proposed method also has the validation set to set its hyperparameters or as AdaAug did in its paper, parts of the training data can be used as validation.

For those reasons, even positive reviewers did not strongly insist on accepting this paper. I would like to encourage to resubmit it to other venue after addressing the above issues especially regarding on experiments.

**Summary Of Ac-Reviewer Meeting:**

As mentioned in the meta review, two points were raised:
- lack of technical novelty
- lack of strong experimental evidence

The second issue was not recognized by the positive reviewers in their original review, but in the meeting, they agreed that the comparisons are possible and necessary, and some of them reduced their scores. In my final decision, the second issue had the most impact, so I suggested rejection.